# Effective engineering of a ketoreductase for the biocatalytic synthesis of an ipatasertib precursor
Sumire Honda Malca[1], Nadine Duss[1], Jasmin Meierhofer[1,4], David Patsch[1], Michael Niklaus [1], Stefanie Reiter[1,5], Steven Paul Hanlon[2], Dennis Wetzl[2,6], Bernd Kuhn[3], Hans Iding [2] & Rebecca Buller [1] ✉

Semi-rational enzyme engineering is a powerful method to develop industrial biocatalysts. Profiting from advances in molecular biology and bioinformatics, semi-rational approaches can effectively accelerate enzyme engineering campaigns. Here, we present the optimization of a ketoreductase from *Sporidiobolus salmonicolor* for the chemo-enzymatic synthesis of ipatasertib, a potent protein kinase B inhibitor. Harnessing the power of mutational scanning and structure-guided rational design, we created a 10-amino acid substituted variant exhibiting a 64-fold higher apparent $k_{cat}$ and improved robustness under process conditions compared to the wild-type enzyme. In addition, the benefit of algorithm-aided enzyme engineering was studied to derive correlations in protein sequence-function data, and it was found that the applied Gaussian processes allowed us to reduce enzyme library size. The final scalable and high performing biocatalytic process yielded the alcohol intermediate with ≥ 98% conversion and a diastereomeric excess of 99.7% (*R,R-trans*) from 100 g L$^{-1}$ ketone after 30 h. Modelling and kinetic studies shed light on the mechanistic factors governing the improved reaction outcome, with mutations T134V, A238K, M242W and Q245S exerting the most beneficial effect on reduction activity towards the target ketone.

Optically pure alcohols are key chiral intermediates in the synthesis of active pharmaceutical ingredients (APIs)[1]. The valuable building blocks are accessible by a host of synthetic methods, including asymmetric hydrogenation, transfer hydrogenation of ketones and (dynamic) kinetic resolution of the alcohol racemates[2–10]. In an industrial context, synthetic strategies, such as the asymmetric reduction of carbonyl compounds, are preferred as they can valorize all starting material satisfying the principles of green chemistry. In particular, the biocatalytic synthesis of optically pure alcohols by ketoreductases (KREDs) has gained considerable attention in the last decade: Ketoreductases can regio- and stereoselectivity transfer a hydride from NAD(P)H to a carbonyl group of the substrate molecule while operating under mild reaction conditions and in the absence of possibly contaminating metal-based catalysts. Notably, recycling of the cofactor on industrial scale has been well established using isopropanol (iPrOH) as cheap hydride donor[11,12]. In addition, the enzymes are produced from renewable resources and are readily biodegradable enabling sustainable industrial processes.

Unlike other families of alcohol dehydrogenases or reductases, KREDs are monomeric proteins that bind the cofactor without a Rossmann-fold motif[13]. Harnessing their exquisite selectivity and high evolvability, KREDs have been applied in the asymmetric reduction of a broad range of ketones and aldehydes to relevant optically pure secondary alcohols[11,14]. In drug development, they have enabled access to chiral synthons of APIs such as cholesterol-lowering atorvastatin[15,16], anti-asthmatic montelukast[17], antiviral simeprevir[18,19], and the anti-cancer ipatasertib[20,21].

Ipatasertib (3) is a potent Akt (protein kinase B) inhibitor API developed for the treatment of metastatic castration-resistant prostate

[1]Institute of Chemistry and Biotechnology, Zurich University of Applied Sciences, Einsiedlerstrasse 31, 8820 Wädenswil, Switzerland. [2]Process Chemistry and Catalysis, F. Hoffmann-La Roche Ltd., Grenzacherstrasse 124, 4070 Basel, Switzerland. [3]Pharmaceutical Research and Early Development, F. Hoffmann-La Roche Ltd., Grenzacherstrasse 124, 4070 Basel, Switzerland. [4]Present address: Analytical Research and Development, MSD Werthenstein BioPharma GmbH, Industrie Nord 1, 6105 Schachen, Switzerland. [5]Present address: Manufacturing Science and Technology, Fisher Clinical Services GmbH, Biotech Innovation Park, 2543 Lengnau, Switzerland. [6]Present address: Nonclinical Drug Development, Boehringer Ingelheim International GmbH, Birkendorfer Strasse 65, 88397 Biberach an der Riss, Germany. ✉e-mail: rebecca.buller@zhaw.ch

cancer and triple-negative metastatic breast cancer[22–24]. The complex molecule is built up in ten synthetic steps and contains three stereocenters that are created by highly selective metal and enzyme catalysis[20,21]. One of these chiral centers is currently introduced by a commercial KRED which is capable of asymmetrically reducing the prochiral ketone **1a** to the desired (R,R)-trans alcohol intermediate **2a** (Fig. 1 and Supplementary Fig. 1). With the aim to illustrate the power of algorithm-aided enzyme engineering in combination with automation for the efficient development of high-performance enzymes, we focused on this crucial reaction step.

As the identification of a robust enzyme starting scaffold is a key prerequisite for the development of a biocatalyst to be used at industrial scale[25–27], we opted to explore a comprehensive KRED toolbox available in our laboratories[28]. In this screen, we selected NADP$^+$-dependent aldehyde reductase II from the red yeast *Sporidiobolus salmonicolor* (UniProt ID: Q9UUN9) for further engineering, herein referred to as *Ssal*-KRED, owing to its absolute stereopreference for the desired alcohol product. *Ssal*-KRED was first isolated and characterized as a selective β-ketoester reductase by Shimizu et al.[29,30] and posterior studies on *Ssal*-KRED and its variants have provided meaningful structural and function data using various alkyl or aryl ketones[31–35].

Building on the initial insights into *Ssal*-KRED's structure-function relationship for the literature-described but much less complex substrates, we aimed to evolve *Ssal*-KRED for the industrial asymmetric reduction of **1a** in which we required the biocatalyst to efficiently perform at high substrate loadings (100 g l$^{-1}$) while operating with an iPrOH-based cofactor recycling system using elevated iPrOH concentrations to drive the reaction equilibrium toward the target product[11]. With these considerations in mind, we embarked on an enzyme engineering campaign targeting increased activity and robustness, understood as enhanced stability, preserved *trans*-diastereoselectivity as well as iPrOH tolerance.

## Results and discussion
### Initial KRED screening
To identify a suitable KRED for reduction of **1a**, we profited from our previously reported in-house KRED collection consisting of 51 active and soluble enzymes of plant, fungal or bacterial origin[28]. To further increase diversity, this panel was supplemented with an additional set of 12 (putative) fungal and bacterial KRED genes, sourced, amongst others, from literature and data mining of the NCBI database (Supplementary Table 1). Screening of the supplemented enzyme collection toward conversion of **1a**, while utilizing a glucose/glucose dehydrogenase-system (Glc/GDH) for cofactor recycling, led to the identification of a KRED from *Sporidiobolus salmonicolor*[36], subsequently named *Ssal*-KRED, as the best candidate. *Ssal*-KRED exhibited promising conversion values (92%) and exquisite diastereoselectivity (99% *de* (*trans*)) at 3 g l$^{-1}$ substrate (Fig. 2 and Supplementary Table 1). Intrigued by these biocatalysis results, we selected *Ssal*-KRED as the starting point for further engineering with the goal of obtaining an industrially viable catalyst to produce the valuable drug intermediate **2a**.

### Identification of "hot spots" through mutational scanning and enzyme modeling
As a first step in the *Ssal*-KRED engineering campaign (Fig. 3), we set out to obtain an overview of amino acid positions which would act as "hot spots" for the transformation of **1a**. For this purpose, single-site saturation mutagenesis (SSM) libraries on every second amino acid position (171 sites) were generated constituting a mutational scanning library (Library 1, L1). Approximately 7700 transformants (76% library coverage) were screened using a UV kinetic assay which followed depletion of **1a** at 340 nm in the presence of KRED-containing *E. coli* lysate, NADP$^+$ and iPrOH (Supplementary Figs. 2 and 3). Calculated using initial rates, fold-improvement over the parent (FIOP) or the wild type (FIOWT) were used as performance indicators of variants. Sequencing of L1 variants with improved performance revealed 27 positions in the substrate environment, second sphere and protein surface as having a positive impact on reductase performance (Supplementary Table 2) and mapped out possible key positions for further engineering. To further guide hot spot selection positions in the direct substrate environment, we modeled **1a** in the available crystal structure of NADPH-bound *Ssal*-KRED (PDB 1Y1P). Informed by the combination of the mutational scanning and in silico docking data, we selected six positions within 4 Å from **1a** located either in the substrate entrance tunnel (F97, A238, L241) or in the substrate cavity (L174, M242, Q245) (Fig. 4a, b) for full randomization in individual SSM libraries (L2). Inclusion of positions M242 and Q245 was additionally supported by literature evidence[32–34]. Sequencing of L2 verified the presence of all 114 possible variants confirming the exhaustiveness of our subsequent activity analysis. The obtained sequence-activity data highlighted that while position 97 exhibited little flexibility in its substitution pattern with tryptophan being the only beneficial mutation (FIOWT = 1.4), the other five sites could be replaced by 4 to 12

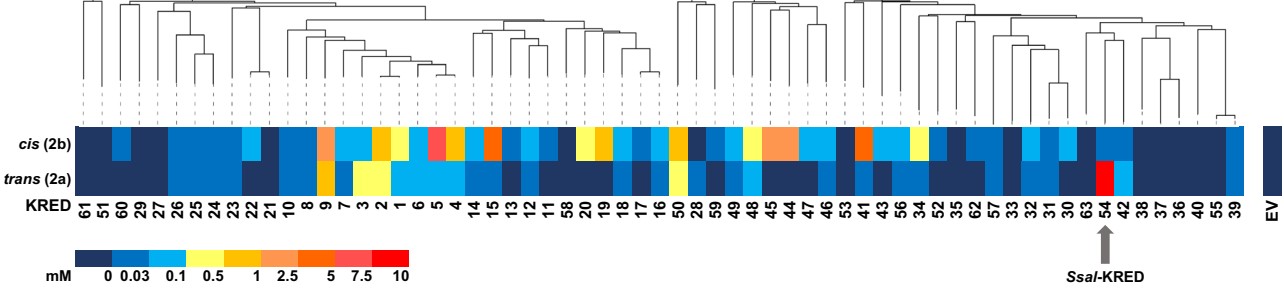

**Fig. 1 | KRED-catalyzed reduction step in the chemo-enzymatic synthesis of ipatasertib.** The prochiral ketone **1a** is stereoselectively reduced to the alcohol **2a**, which is in turn utilized as a precursor of ipatasertib (**3**).

**Fig. 2 | Screening of an in-house collection of 63 KRED wild-type enzymes toward 1a reduction.** Ketone **1a** (c = 3 g l$^{-1}$ equivalent to 10 mM) was reduced in the presence of a glucose/glucose dehydrogenase system for NAD(P)H regeneration. KREDs are ordered based on their phylogenetic relationship (iTOL v5, EMBL)[65].

Concentrations of target product **2a** and its diastereomer **2b** ((R,S)-*cis* alcohol) are indicated. KRED 54 (*Ssal*-KRED) yielded the highest conversion and *trans*-diastereoselectivity within the collection. EV empty vector (negative control). Source data for this figure is available (Supplementary Data 2).

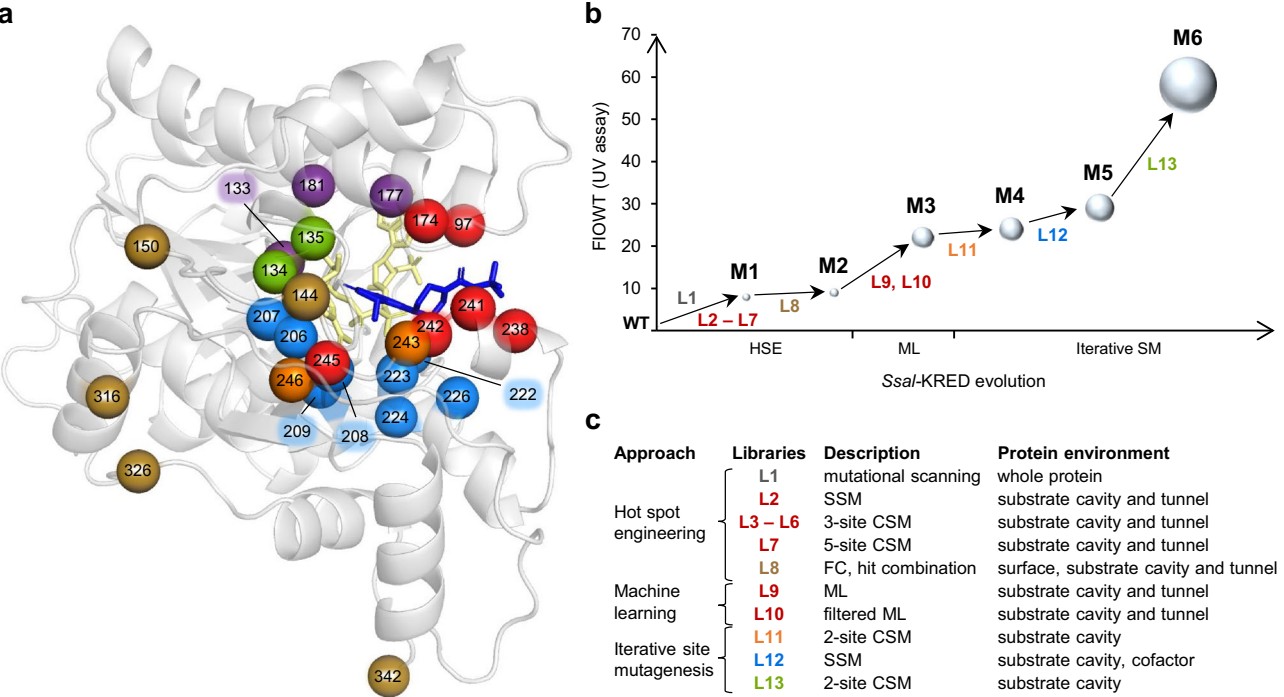

**Fig. 3 | Engineering of *Ssal*-KRED for efficient reduction of 1a. a** Binding model of **1a** (blue) in the active pocket of NADPH-bound (yellow) *Ssal*-KRED (PDB ID: 1Y1P). Catalytic residues are indicated in purple. Targeted positions (23) are color-coded in accordance with the library they were investigated in (for reference, see **c**). Amino acid positions covered in L1 (mutational scanning library) are not depicted. **b** Overview of the relative activity of the *Ssal*-KRED hit variants **M1**–**M6** compared to the wild-type enzyme. The fold-improvement over the wild type (FIOWT) was

determined using an UV assay (Supplementary Fig. 3). **c** Libraries (L1–L13) on protein surface, substrate entrance tunnel, substrate cavity and cofactor environment positions were built using single-site saturation mutagenesis (SSM), combinatorial saturation mutagenesis (CSM) or focused combinatorial (FC) mutagenesis, integrating hit combinations and combinatorial libraries. L10 is a filtered ML-library, which was designed by combining knowledge from previous rounds of engineering with ML predictions.

different amino acids and still achieve wild-type or even improved reductase performance (Supplementary Table 3). Overall, single-site variants M242F and Q245T performed best in the UV-assay exhibiting FIOWT of 2.6 and 3.6, respectively. However, variant Q245T lost absolute *trans* preference (99.7% *de*) (Supplementary Fig. 4a), evidencing the role of this position in diastereoselectivity, as previously described with other substrates[33,34].

**Machine learning-aided enzyme engineering**

With the goal to explore large parts of sequence space in silico and equipped with knowledge about the malleability of key residues in the active site, we set out to design combinatorial libraries which would allow to generate a rich data set consisting of sequence and activity information for machine learning (ML) predictions. Considering that tryptophan was the only beneficial mutation at position F97, we used the single variant F97W as the parental enzyme and strategically grouped the five selected residues into four 3-site (L3–L6) and one 5-site (L7) combinatorial saturation mutagenesis (CSM) libraries. Due to time- and resource restriction, we opted to limit our efforts to four out of all possible 3-site combinatorial libraries. The library design was guided by geometrical considerations leading us to group amino acid positions most likely to interact (Fig. 4b). To limit library diversity to the theoretical number of variants (e.g., $20^5$ in the 5-site combinatorial library), we opted for an "one codon encoding one amino acid" gene library ordered from Twist Biosciences. The fragments were cloned into the pET22b(+) vector and transformed into *E. coli* BL21 (DE3) cells in house. Using Sanger sequencing, we confirmed 2133 and 762 unique on-target variants for the 3-site libraries and the 5-site library, respectively, corresponding to a library coverage ranging from 5.5–8.5% for libraries L3–L6 and 0.024% for library L7. Within this design space, library L4 interrogating residues L241, M242 and Q245 yielded best-performing variants (Supplementary Tables 4 and 5). The best L4 variant **M1**

(F97W_L241M_M242W_Q245S) exhibited a FIOWT of 8 while preserving absolute *trans*-diastereoselectivity when assayed under optimized conditions (Fig. 5; Supplementary Fig. 4b and Supplementary Table 6).

Going forward, we set out to explore the remaining protein landscape in silico using Gaussian processes. In analogy to a previous successful application of machine-learning for enzyme design in our laboratory[37], we employed a strategy in which we represented amino acids by their different physicochemical and biochemical characteristics, derived from the AAin-dex database[38,39] (see "Methods" section). Each enzyme sequence was then represented by a feature vector which was defined by joining the vector representations of its individual amino acids at the selected library sites, e.g., in case of the 5-site combinatorial library at L174X, A238X, L241X, M242X and Q245X.

It is important to note that within the budding field of algorithm-aided enzyme engineering, only few examples exist which showcase the successful application of machine learning to improve enzyme activity, the unarguably most complex enzymatic function[37,40–44]. In addition, the extent of sampling to obtain predictions for activity or other protein characteristics varies strongly[37]. Yet, to effectively reduce the physical screening burden in enzyme engineering campaigns, it is essential to have guidelines that define the degree to which a variant library needs to be experimentally screened. To address this gap in knowledge—at least in part—we decided to carry out our machine-learning based predictions in two stages. First, we used the sequence-function data of the best-performing library L4 (651 data points; 8% coverage) as the basis to make Gaussian process-based ML predictions. Notably, quality control of L4 had shown that in this library all 20 amino acids were present at each randomized position leading to a good distribution of screened variants over the entire protein landscape (Supplementary Fig. 5). Evaluating the results of the ML-analysis, we constructed library L9 consisting of the best 24 ML-predicted variants in addition to six randomly chosen variants with a lower ranking for exploratory purposes

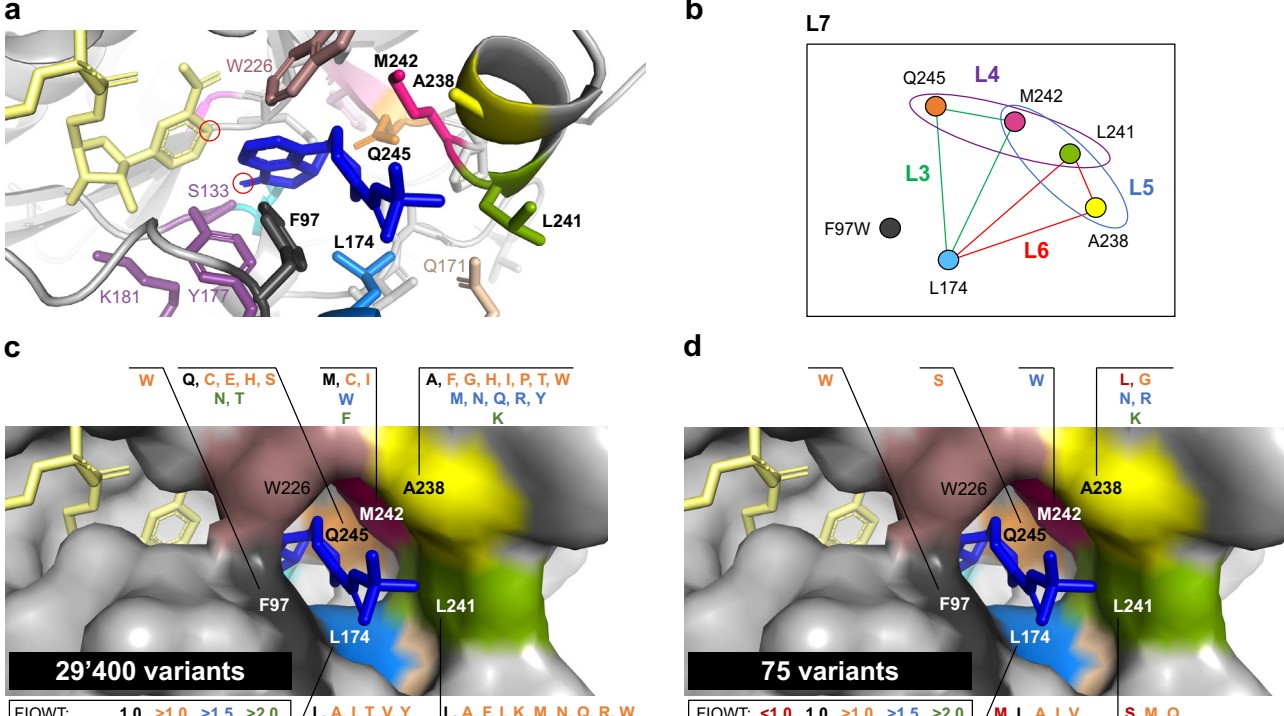

**Fig. 4 | Substrate environment residues targeted in the hot spot engineering and machine learning approaches. a** Model of *Ssal*-KRED binding **1a** (blue) and NADPH (yellow). The targeted keto moiety in **1a** and the hydride-donor C4 of the nicotinamide ring of the cofactor are circled in red. Catalytic residues are depicted as purple sticks. The six positions targeted in SSM libraries L2 are indicated in bold letters: F97, L174, A238, L241, M242, Q245. **b** Schematic representation of hot spot grouping for CSM libraries. While keeping mutation F97W fixed, five selected residues were grouped into four 3-site CSM libraries designated as L3 (174-242-245), L4 (241-242-245), L5 (238-241-242) and L6 (174-238-241). In addition, all five residues were simultaneously saturated in a 5-site CSM library (L7). **c** Surface representation depicting neutral to beneficial mutations (color-coded according to their FIOWT values) in the context of the wild-type enzyme. Linearly combining these mutations would have led to a theoretical library size of 29,400 variants. **d** Sequence-activity data of L3–L7 were used to train a ML algorithm. Predicted variants were combined in a small library of 75 variants (L10). Substitutions, which had not been found beneficial in context of the wild-type enzyme (L2) but were predicted to perform well in the ML-filtered library L10, are highlighted in red.

| Library | Approach | Variant | 97 | 134 | 224 | 238 | 241 | 242 | 245 | 246 | 316 | 342 | FIOP | FIOWT |
|---|---|---|---|---|---|---|---|---|---|---|---|---|---|---|
| | | | | | | Substrate / Cofactor | | | | | Surface | | | |
| – | – | WT | F | T | S | A | L | M | Q | Y | L | T | 1.0 | 1 |
| L2 – L7 | HSE | M1 | W | T | S | A | M | W | S | Y | L | T | 8.0 | 8 |
| L8 | HSE | M2 | W | T | S | A | M | W | S | Y | M | M | 1.1 | 9 |
| L9, L10 | ML | M3 | W | T | S | K | M | W | S | Y | M | M | 2.5 | 22 |
| L11 | ISM | M4 | W | T | S | K | M | W | S | G | M | M | 1.1 | 24 |
| L12 | ISM | M5 | W | T | A | K | M | W | S | G | M | M | 1.2 | 29 |
| L13 | ISM | M6 | W | V | A | K | M | W | S | G | M | M | 2.0 | 58 |

**Fig. 5 | Mutations and fold-improvement of *Ssal*-KRED hit variants.** *Ssal*-KRED hit variants were obtained via hot spot engineering (HSE), machine learning (ML) and iterative site mutagenesis (ISM). Positions are color-shadowed in accordance with the library they were investigated in (for reference, see Fig. 3a, c). The fold-improvement over the wild type (FIOWT) was calculated from fold-improvement over the parent (FIOP) values. AA amino acid, − not applicable.

(Supplementary Table 7, https://github.com/ccbiozhaw/Ssal-KRED_evolution). Disappointingly, the best variants of the ML-based library L9 displayed FIOWT values in the range of known variant **M1** (Supplementary Table 8) indicating that either our experimental screening had already serendipitously identified the best-performing variant within the design space, or, that the supplied data set had not been sufficient for our ML-algorithm to identify more successful enzyme solutions.

To explore a situation in which it would be highly unlikely that the best sequence-function solution had already been measured experimentally, we opted to explore the quality of predictions when the ML-algorithm was supplied with a data set that covered the design space less exhaustively. For this purpose, we used the sequence-activity data generated in the frame of three 3-site combinatorial libraries with an $R^2$ score >0.6 (L3, L4 and L6), the five-site library L7, as well as the data from library L9 to train the ML algorithm again (2453 datapoints; 0.08% library coverage, https://github.com/ccbiozhaw/Ssal-KRED_evolution) (Supplementary Fig. 6). As a parental scaffold for the second ML library, we used variant **M2** (F97W_L241M_M242W_Q245S_L316M_T342M), in which the best activity-engineered variant **M1** had been combined with the most beneficial surface residue mutations identified via the mutational scanning library L1 (Library L8, Supplementary Tables 9 and 10). Variant **M2** exhibited a FIOWT of 9 (Supplementary Table 11) as well as a thermal and organic solvent stability comparable to that of the wild type (Supplementary Table 12). We opted to follow a different approach than in our previous ML evolution round: Instead of ordering distinct enzyme sequences, we constructed a ML-filtered library L10 consisting of 75 variants, in which substrate binding sites L174, A238 and L241 were modulated to a small set of predicted amino acids while M242 and Q245 were fixed to tryptophan and serine, respectively (Fig. 4d, Supplementary Table 13 and Supplementary Fig. 7). For this "filtered" library, we opted to supplement the machine learning predictions with knowledge from previous engineering rounds, expanding or reducing the employed amino acid alphabet at each of the modulated positions rationally (Supplementary Table 13). As anticipated, the construction of this library allowed us to effectively reduce sequence space. Compared to a library combining all beneficial mutations identified

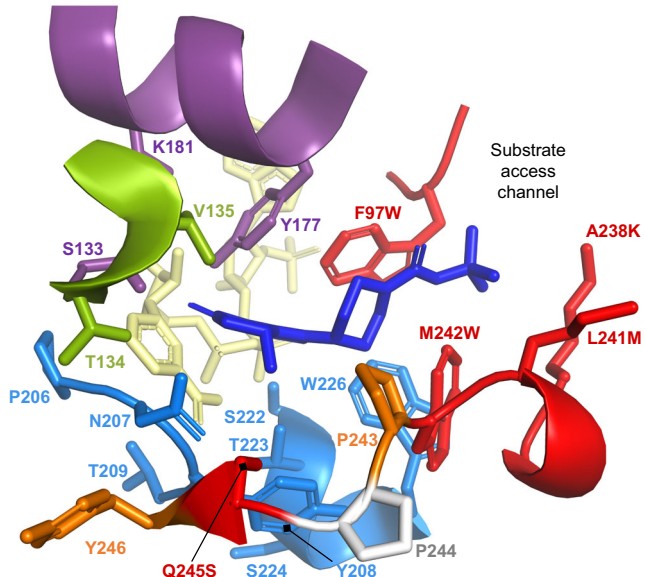

**Fig. 6 | Residues targeted in the iterative site mutagenesis approach.** Homology model of **M3** (F97W_A238K_L241M_M242W_Q245S_L316M_T342M) with bound **1a** (blue) and cofactor (yellow). Catalytic residues are depicted in purple. Mutated residues in the active site and tunnel entrance are displayed in red, surface mutations L316M and T342M are not shown. Positions in the substrate or cofactor environments targeted for the iterative site mutagenesis libraries are colored in accordance with the libraries they are located in: 2-site CSM library L11 (orange), SSM libraries L12 (marine blue) and 2-site CSM library L13 (green).

in L2 (theoretical library size of 29,400 variants; Fig. 4c) or a fully randomized 5-site library (theoretical library size of $20^5$ variants), we could decrease the screening burden by a factor of at least 392 or even $4.2*10^4$, respectively. It should also be noted that improved variants stemming from library L10 contained amino acid substitutions which had not been beneficial in the context of the wild-type enzyme (Supplementary Table 3), for example L174M, A238L, and L241S. As before, library completeness was confirmed via Sanger sequencing before experimental analysis. Screening of library L10 led to the identification of variant **M3** (F97W_A238K_L241M_M242W_Q245S_L316M_T342M), which displayed a FIOWT of 22, corresponding to a 2.5-fold improvement over parent M2, and preserved absolute *trans*-diastereoselectivity (Fig. 5). Concomitant substitutions A238G/R/L_L241M and A238K_L241Q/S also proved beneficial as variants carrying these mutations showed a FIOWT of 12–17 (Supplementary Table 14).

Clearly, the effectiveness of the approach was influenced by the decision to construct an enzyme library, which was informed by machine learning predictions and knowledge about previous successful variants, instead of solely testing a ranked set of predicted sequences. In this context, potential uncertainties of the model were compensated by maximizing the probability via greedy exploitation of promising predicted mutations. An alternative approach to address potential bias introduced by limited and imbalanced training datasets would have been to follow an iterative approach of balancing exploration and exploitation through Bayesian learning techniques[45].

Finally, to assess the quality of L10, we compared the ratio between positive (FIOWT ≥ 1) and negative variants (FIOWT < 1). While library L2 to L7 predominantly displayed negative variants (ranging from 66% to 97%), the filtered ML library L10 enabled the enrichment of positive variants (76%; Supplementary Fig. 8). As we provided all collected data (e.g., sequence-function data of positive and negative variants) to train the Gaussian process and subsequently filtered with prior knowledge, it is unclear if the increase in the fraction of hit enzymes was achieved by learning to build on the positive combinations or by learning to avoid the unfavorable combinations, or both. In this sense, we recommend that all

data points collected under comparable experimental conditions, including data connected to negative variants, are provided to improve on the predictive ability of machine learning models.

## Iterative site mutagenesis

Besides high turnover rates, industrial biocatalysts must perform efficiently also at increased substrate loads[46]. Cognizant of the fact that commercial production of **2a** was targeted to take place at a substrate load of $100 \text{ g l}^{-1}$ and at a defined iPrOH concentration of 8% (v/v), we evaluated the effect of time and temperature on conversion levels achieved by **M1**, **M2** and **M3** (Supplementary Fig. 9 and Supplementary Data 3). Using 0.2 ml-biocatalytic reactions at $100 \text{ g l}^{-1}$ substrate, we could show that variant **M3** was able to reach a promising 70% substrate conversion at 28 °C after 24 h. To further increase M3's catalytic prowess we decided to employ iterative site mutagenesis[47] on several residues in or close to the active site (Fig. 6).

As a first step to increase the enzyme's reduction activity toward **1a** further, we probed positions 243 and 246, flanking the critical S245 residue, by constructing a 2-site combinatorial site mutagenesis library (L11). Screening this library revealed variant **M4** (F97W_A238K_L241M_M242W_Q245S_Y246G_L316M_T342M) as the most successful biocatalyst with an apparent FIOWT of 24 (Fig. 5 and Supplementary Table 15). Importantly, this variant reached 78% conversion to **2a** after 24 h in reactions at $100 \text{ g l}^{-1}$ substrate (Supplementary Fig. 10 and Supplementary Data 4), leading us to continue our evolution campaign along this lineage.

Interrogation of an additional eight substrate- or NADP(H)-binding residues (P206, N207, Y208, T209, S222, T223, S224, W226) in the frame of SSM libraries (L12) using variant **M4** as parent showed that this sequence space did not harbor many improved enzyme solutions for reduction of **1a** (Supplementary Table 16). While position 207, 209, 222 and 223 revealed neutral changes at best, any amino acid substitution on positions 206 and 208 exhibited a deleterious effect on reductase performance as evidenced by high number of inactive or unimproved (FIOP ≤ 1) variants. Only exploring residue S224 led to an improved reductase, named **M5** (F97W_S224A_A238K_L241M_M242W_Q245S_Y246G_L316M_T342M), which exhibited a FIOWT of 29 (Fig. 5). This variant also displayed higher tolerance to heat treatment than any previously activity-engineered variant (Supplementary Table 17).

To conclude the evolution campaign, we elected to interrogate hot spots T134 and V135, adjacent to the key catalytic residue S133. Our selection was guided by a previous study, in which simultaneous randomization of these residues in enzyme *Ssal*-KRED had led to variant with a 4.6-fold improved turnover rate compared to the wild type for the reduction of a disubstituted cyclic diketone[35]. When screening the final 2-site CSM library (L13) in context of the **M5** scaffold, we identified variant **M6** (F97W_T134V_S224A_A238K_L241M_M242W_Q245S_Y246G_L316M_T342M) displaying a FIOWT of 58 over the wild type (Fig. 5 and Supplementary Table 18) allowing us to meet the activity-improvement goal of the engineering study.

## Preparative biocatalytic reactions

Conclusively, the performance of the hit variants was assessed in 1 ml-scale biocatalytic reactions at $100 \text{ g l}^{-1}$ substrate (100 mg of **1a**) with lyophilized *E. coli* cell lysates harboring the appropriate biocatalysts at 30 °C (Supplementary Fig. 11 and Supplementary Data 5). Gratifyingly, we found that our most active variant **M6** performed well in the final formulation and under the industrial conditions and allowed us to synthesize **2a** with 96% conversion and >99.5% *de* (*trans*) in 24 h without the need to remove acetone. While all variants followed the activity trends obtained in the kinetic measurements (*vide infra*), it should be noted that **M5** performed less well than its parent enzyme **M4** when formulated as a lyophilized cell lysate.

Based on the evaluation at 1 ml scale, variant **M6** was selected for further performance analysis and upscaling to 100 ml-scale, corresponding to $100 \text{ g l}^{-1}$ of **1a** in the biocatalytic reactions. To our delight, the engineered biocatalyst could convert 10 g of **1a** with ≥98% conversion in 30 h, generating **2a** with 99.7% *de* (*R,R-trans*) when applying the iPrOH system and

## Table 1 | Comparison of KRED-mediated syntheses of 2a

| Parameter | *Ssal*-KRED (WT) | *Ssal*-KRED_M6 | |
|---|---|---|---|
| s/e[a] | 5 | 5 | 50 |
| KRED[b] | 20 mg | 2 g | 0.2 g |
| CRS[c] | iPrOH | iPrOH | Glc/GDH |
| Substrate 1a | 100 mg | 10 g | 10 g |
| Conversion[d] | 26%[e] | 99% | >99% |
| *de* (*trans*) | >99.5% | 99.7% | 99.9% |
| HPLC-purity[f] | n.d. | 99.7% | 99.5% |
| Reaction time | 24 h | 30 h | 27 h |
| Temperature | 23 °C | 23 °C | 23 °C |

*n.d.* not determined, *WT* wild-type enzyme.
[a]Substrate-to-enzyme ratio.
[b]As lyophilized enzyme lysate.
[c]Cofactor recycling system.
[d]Determined by achiral HPLC.
[e]Reaction stalled.
[f]Determined by chiral HPLC.

## Table 2 | Kinetic characterization of selected *Ssal*-KRED variants using ketone 1a for the synthesis of 2a

| *Ssal*-KRED variant | app. $K_M$, [µM] | app. $k_{cat}$, [min$^{-1}$] | app. $k_{cat}/K_M$, [min$^{-1}$ mM$^{-1}$] | rel. $k_{cat}$ |
|---|---|---|---|---|
| WT | 11.9 ± 1.1 | 0.49 ± 0.02 | 41 | 1 |
| M1 | 162.8 ± 7.8 | 5.8 ± 0.1 | 35.5 | 12 |
| M2 | 175.2 ± 9.9 | 6.2 ± 0.2 | 35.6 | 13 |
| M3 | 299.7 ± 7.9 | 15.3 ± 0.4 | 51.2 | 31 |
| M4[a] | 1668 ± 103 | 45.8 ± 2.4 | 27.5 | 94 |
| M5 | 471.7 ± 34.6 | 21.3 ± 1.0 | 45.2 | 43 |
| M6 | 143.5 ± 5.5 | 31.5 ± 0.8 | 219.7 | 64 |

Source data for this table is available (Supplementary Data 6).
*WT* wild-type enzyme.
[a]Substrate saturation was not reached.

in a substrate-to-enzyme (s/e) ratio of 5 (Table 1; Supplementary Figs. 12–15 and Supplementary Data 1). When using a Glc/GDH cofactor regeneration system, it was additionally possible to reduce the amount of enzyme by 10-fold (s/e 50) while obtaining similar results in terms of conversion and target product purity (Table 1 and Supplementary Figs. 16 and 17). Overall, both settings, e.g., the use of Glc/GDH or iPrOH for cofactor regeneration, offered technical feasible and commercially viable applicability.

### Enzyme kinetics

To gain further insights into the achieved activity improvements, we opted to purify key enzyme variants along the evolution trajectory (Fig. 5 and Supplementary Fig. 18) and measure the apparent Michaelis-Menten parameters when iPrOH was used as the sacrificial cofactor. In a first step, we compared the UV-assay designed to record the consumption of NADPH with a HPLC-method quantifying product formation by measuring the wild-type enzyme with both approaches (UV-assay: $k_{cat}$ = 0.49 ± 0.02 min$^{-1}$; $K_M$ = 11.9 ± 1.1 µM; HPLC-assay $k_{cat}$ = 0.49 ± 0.02 min$^{-1}$; $K_M$ = 11.5 ± 0.5 µM; Supplementary Fig. 19 and Supplementary Data 6). Based on the good agreement of the obtained Michaelis-Menten parameters, we decided to use the UV method for the analysis of further *Ssal*-KRED variants due to its intrinsic ease-of-use.

For all enzymes, the Michaelis-Menten analyses showed that the $k_{cat}$ values of the engineered reductases were steadily increasing along the evolutionary trajectory and were directly in line with the FIOWT values generated by applying clarified cell lysates in the UV assay (Table 2). This trend evidenced that the specific enzyme activity was improved through our engineering campaign rather than enzyme expression. The most active variant **M6** to emerge after 13 individual libraries exhibited a 64-fold improved apparent $k_{cat}$ when compared to the wild-type enzyme (**M6** $k_{cat}$ = 31.5 ± 0.08 min$^{-1}$ vs. wild type $k_{cat}$ = 0.487 ± 0.018 min$^{-1}$) while the apparent $K_M$ was found to be ~12-fold higher than for the wild-type enzyme (**M6** $K_M$ = 143.5 ± 5.5 µM vs. wild type $K_M$ = 11.9 ± 1.1 µM).

Given the importance of high turnover numbers for industrial biocatalysis, where rapid conversion of substrate to product is required and the substrate load is typically very high, this is a particularly desirable evolution outcome. In this context it is noteworthy that for variant **M4** substrate saturation could not be reached in the kinetic experiments indicating an elevated $K_M$ compared to all other *Ssal*-KRED variants (Supplementary Fig. 19). However, we found that the estimated $k_{cat}$ was particularly high (45.8 ± 2.4 min$^{-1}$; 94-fold improvement over wild type). In line with these results, variant **M4** was also found to perform well when tested in 1 ml-scale reactions at 100 g l$^{-1}$ substrate (Supplementary Fig. 10).

### Enzyme modeling

To map the observed improved protein characteristics to structure, we modeled key hit variants **M1**, **M3** and **M6**. In a first step, we investigated mutations common to all three variants (F97W, L241M, M242W and Q245S) and analyzed their structural effects compared to the wild-type enzyme. Interestingly, mutations F97W and M242W seem to generate a hydrophobic cleft around the ring of the *N*-Boc piperazine moiety of **1a**. Moreover, together with W226 located at the top of the substrate cavity, they appear to form a unique aromatic cage-like binding site (Fig. 7a). Although substrate affinity decreased in these variants in comparison with the wild type (Table 2), this cage might enable a more favorable placement of **1a** with respect to catalytic residues. Mutation L241M could contribute in a similar way by influencing adjacent M242W. Key mutation Q245S, which lies at the back of the binding pocket and in proximity to the methyl substituent of the cyclopentyl moiety of **1a**—opposite to the targeted carbonyl group—is likely necessary to enlarge the binding pocket (Fig. 7b) improving placement of **1a** for catalysis. In support of this theory, we found that incorporation of the slightly larger threonine at position 245 seems to change positioning of the target carbonyl group as evidenced by a drop in selectivity (variant Q245T, Supplementary Table 8). Moreover, mutation Q245T caused a decreased reaction rate in the context of more evolved variants (e.g., in combination with F97W_L241Q/A_M242W, a FIOWT of 7–8 was obtained in the presence of Q245S while the corresponding Q245T variants displayed a lower FIOWT of 3–4, Supplementary Table 8).

The final variant **M6** harbors the additional mutation T134V, which lies in proximity to catalytic residue S133. In previous studies amino acid T134 has been described to anchor functional groups in other small molecules, e.g., the non-targeted carbonyl oxygen atom in the cyclic diketone camphorquinone or the chlorine atom in ethyl 4-chloroacetoacetate[31]. In case of substrate **1a**, we observed that substituting T134 with a hydrophobic residue was beneficial presumably because **1a** lacks other functional groups near the target carbonyl group. Furthermore, we observed that because of the T134V mutation polar interactions between the side chain of position 134 and NADPH-binding residues P206 and N207 were lost. The rearrangement of polar interactions in the active site might allow for a more efficient interaction of residues P206 and N207 with the cofactor NADPH (Fig. 7c) in this way positively affecting activity[48].

Along the evolutionary trajectory, secondary structure features changed particularly due to mutations located in the substrate entrance tunnel, namely F97W (loop region 91–101), A238K (helix α8) and L241M (loop region 241–245). A238K does not seem to interact with neighboring residues or the positioned substrate, yet it has a large effect on turnover. Given the literature-described properties of lysine, it might act as a "tunnel gate" (Fig. 7d), involving a side chain conformational change and in this way modulating access of the substrate or the solvent into the tunnel[49]. Mutation Y246G, present only in **M6**, appears to cause β-strand 3 slightly move toward the substrate, thus potentially influencing the rotamer orientation of

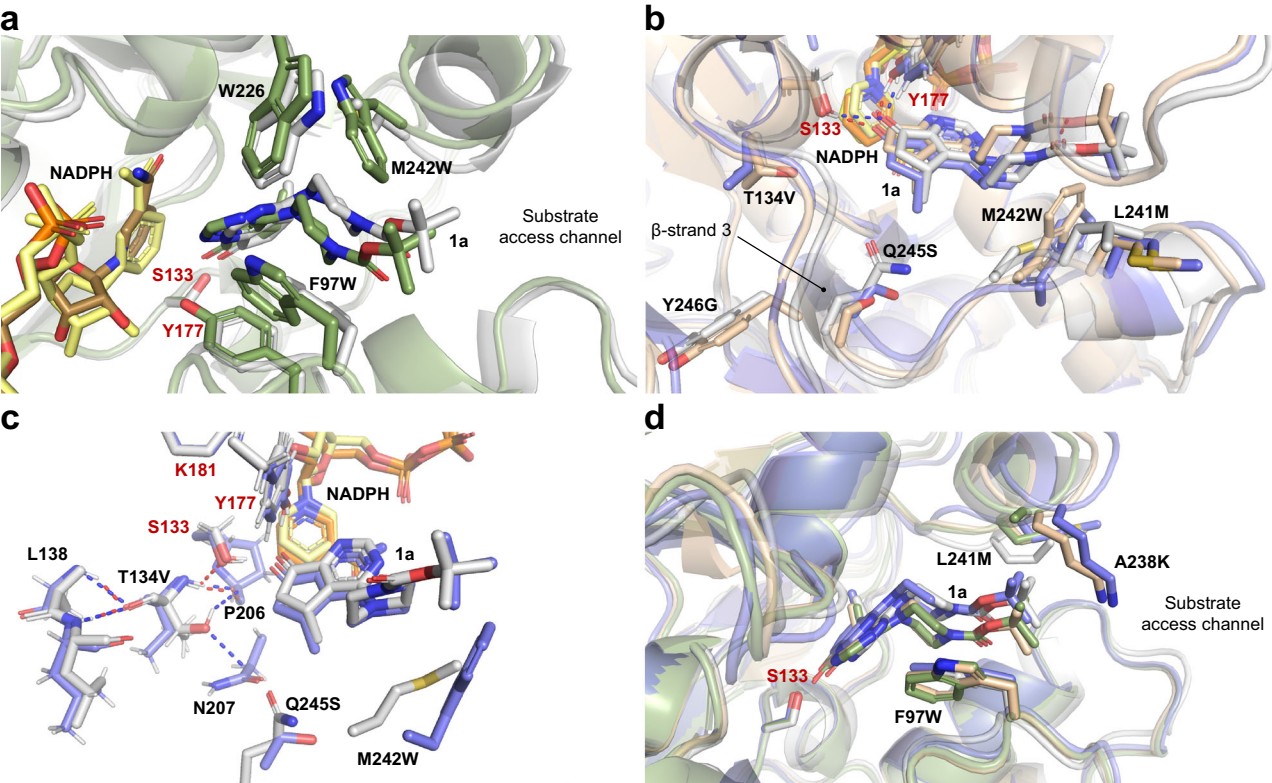

**Fig. 7 | Modeling of key variants M1, M3, and M6.** Homology models of wild-type *Ssal*-KRED (light gray) and variants **M1** (smudge), **M3** (wheat) and **M6** (slate) with binding modes of **1a**. The cofactor is shown in pale yellow, sand, yellow and orange, respectively. Catalytic residues are indicated in red letters. **a** Superimposed WT and **M1** displaying the aromatic cage-like binding site formed by F97W, M242W and W226 in the latter (only one variant is shown for clarity). **b** Close-up view on key positions 241, 242 and 245 in the WT, **M3** and **M6**. Coordination of **1a** by catalytic

S133 and Y177 is indicated in blue and red dashes for the WT and **M6**, respectively. **c** Polar interactions of key position 134 with other residues in the WT (blue dashes) and **M6** (red dashes). Interactions with P206 and N207 are lost in variant **M6**. **d** Secondary structure differences on positions 97, 238 and 241 between WT and all evolved variants. Mutation A238K in **M3** and **M6** might act as a "tunnel gate" residue.

Q245S or its proximity to the substrate (Fig. 7b). Looking forward, more detailed information on the structural features governing catalysis could be obtained through X-ray crystallography studies of the cofactor- and substrate-bound enzyme variants.

## Conclusion

In our enzyme evolution campaign, we found that an engineering strategy consisting of enzyme hot spot identification through mutational scanning libraries followed by the set-up of structure-guided libraries, usable to train machine learning algorithms, allowed us to effectively develop a high-performance, industrial biocatalyst. Within six evolution rounds we created a 10-amino acid substituted variant, which exhibited a 64-fold higher $k_{cat}$ and improved robustness under process conditions. While algorithm-based approaches have been shown several times to successfully identify improved enzyme variants in unexplored parts of sequence space[45,50], other examples report that the machine learning effect could be traced back to simple linear additivity arising from combining positive individual mutations[42,51]. When analyzing the composition of our best variant **M6** (F97W_T134V_S224A_A238K_L241M_M242W_Q245S_Y246G_L316M _T342M) in this light, our machine learning algorithm seems—at least at first glance—to also simply have helped us find linearly additive mutations. However, "simply" combining all identified beneficial and wild-type residues from L2 (L174: 6 residues; A238: 14 residues; L241: 10 residues; M242: 5 residues; Q245: 7 residues) in a combinatorial fashion would have led to a library size of 29,400, which, when considering the necessary oversampling and inclusion of controls, would have required us to screen close to a thousand 96-well plates. Filtering machine learning predictions with data from previous engineering rounds supported the construction of a library

consisting of only 75 variants, which reduced the screening burden considerably. Notably, the library also harbored improved enzyme solutions containing for example substitution A238G or A238L, which were neutral or deleterious mutations within the context of the wild-type enzyme (L2) and would thus have been difficult to find. Nevertheless, it should be noted that the benefits of reducing the library size comes at the cost of previously having to screen and sequence variants for the machine learning algorithm (in our case 2453 datapoints). Such an endeavor might thus be most profitable if the sequence-function pairs generated to train the algorithms stem from libraries which are substrate "naïve", e.g., have been constructed in the frame of an early enzyme engineering round. In this way, the expensive sequence data can be reused for future (machine-learning)-aided enzyme engineering campaigns dedicated to other substrates.

Next to Gaussian processes, which were applied here, various other ML algorithms can be employed to predict protein characteristics, such as solubility, stability, specificity, or activity. These methodologies, including partial least-squares regression, random forest, decision trees, support vector machines, K-nearest neighbors, and neural networks, have been discussed in comprehensive reviews[52–57]. Overall, the machine-learning based evolution round was one puzzle piece in a comprehensive enzyme optimization campaign: We further improved the enzyme activity and stability by targeting experimentally determined and literature-derived[31,33–35,58] amino acid positions by applying iterative site saturation mutagenesis[47]. Overall, key mutations for the efficient reduction of **1a** as elucidated by kinetic studies, and modeling included active site residues T134V, A238K, M242W and Q245S while changes on the enzyme surface and in the substrate access tunnel more likely contribute to stability of the enzyme under process conditions.

At the end of the evolution campaign for *Ssal*-KRED covering 180 amino acid positions in 13 libraries (Supplementary Tables 19 and 20), we used our best engineered KRED in preparative scale reactions showcasing a technical feasible and commercially viable process of ipatasertib intermediate **2a**. The developed biocatalytic process is characterized by remarkably high product yield and purity level.

## Methods

### General information
All chemicals, reagents and kits were purchased from commercial suppliers (Supplementary Method 1.1). Substrate **1a** as well as products **2a** and **2b** were synthesized internally (Supplementary Method 1.2). For NMR spectra see Supplementary Data 1. NMR spectra were recorded on a Bruker Avance III 600 MHz spectrometer using TopSpin (4.x) as software. ACD/NMR Workbook 2019 was utilized for the analysis of NMR data.

### KRED cloning and expression
Genes from the in-house KRED collection as well as *Ssal*-KRED libraries L1 and L7 were provided by Twist Bioscience (San Francisco, CA, USA). Other libraries were constructed by PCR amplification using ratio-tuned NDT/VHG/TGG primers (22c-trick technique)[59], NNK primers or other specific (non-degenerate or manually-mixed degenerate) mutagenic primers on defined variant templates, followed by overlap extension PCR with general flanking primers for In-Fusion cloning or T5-exonuclease-dependent assembly (TEDA)[60]. Library L8 and hit combinations were generated using the MEGAWHOP PCR technique[61] with mutagenic primers on the wild type or defined variants. *Ssal*-KRED wild-type and hit variants M1–M6 were subcloned into pET28b(+) for the insertion of an *N*-terminal 6xHis-tag. pET22b(+) and *E. coli* BL21(DE3) were used for all libraries as vector and expression strain, respectively. See Supplementary Method 1.3 for full details. All oligonucleotide sequences used for cloning are provided in Supplementary Table 21.

### KRED expression and lysate preparation
Conditions for cultivation, KRED expression, cell lysis, lysate lyophilization and protein purification are described in Supplementary Methods 1.4 and 1.5. Fresh clarified lysates were used for screening via a UV-based activity assay or micro-scale biocatalytic reactions. On occasion, KRED variants were subjected to heat or organic solvent treatment prior to the UV assay (Supplementary Method 1.6).

### UV assay-based screening
Reductase activity was measured in 96-well Greiner microtiter plates using a spectrophotometer (SpectraMAX Plus, Tecan Spark or Tecan Infinite M Nano+) and corresponding software (SoftMax Pro 4.7.1 or SparkControl 2.1). Reactions were performed in a total volume of 200 μl, with components added in the following order: (1) 178 μl of 0.1 M potassium phosphate buffer pH 7 with 2 mM MgCl$_2$ and 0.01 mg ml$^{-1}$ NADP$^+$ (as sodium salt); (2) 6 μl of clarified lysate (pure or diluted in 0.1 M potassium phosphate buffer pH 7 + 2 mM MgCl$_2$, for a final lysate concentration of 3, 1.5, 1, 0.5 or 0.25% (v/v), according to the library); and (3) 16 μl of a stock solution containing 1.25 mg ml$^{-1}$ **1a** in iPrOH. The assay was run with orbital shaking at 432 rpm and at temperatures oscillating between 28 and 32 °C (room temperature plus 5 °C caused by shaking). Depletion of **1a** was followed at 340 nm and recorded every 5 min for 80 min. Slopes (ΔA min$^{-1}$) within the linear range were used for fold-increase over the parent (FIOP) calculations. FIOP = initial reaction rate of variant ÷ initial reaction rate of parent.

### Machine learning
Amino acid sequence and activity data (FIOP values) analyses of library variants were used as input for machine learning (ML) which was adapted from Buechler et al.[37] The target label (activity) was calculated by dividing the slope of each sample on a plate by the averaged wild-type slope for that plate. The individual variants were represented based on the amino acids'

various physicochemical and biochemical properties at the mutated sites. These properties were derived from the AAindex database[38,39], an extensive collection of amino acid characteristics from several sources. Thirteen features per site, i.e., 13 × 3 (241-242-245 for L9) and 13 × 5 (174-248-241-242-245 for L10), were considered. We then defined the feature vector of a sequence by joining the vector representation of its individual amino acids at the defined sites and aggregated them into a 651 × 39- and a 2453 × 65-dimensional training matrix for L9 and L10, respectively. All predictions were made based on the Algorithm 2.1 of Gaussian Processes for ML (GPML) by Rasmussen and Williams[62], implemented in the scikit-learn Python module. To mitigate overfitting and enhance the assessment of the generalizability of our model, we cross-validated over ten splits, and model performance was evaluated using the coefficient of determination ($R^2$). As a result, average $R^2$ scores of 0.66 and 0.77 were achieved for L9 and L10, respectively (compare predicted vs. measured, Supplementary Fig. 6). Plotly (5.x) and Python (3.8.x) were employed for data visualization. The code, data, and related Supplementary Information can be accessed at https://github.com/ccbiozhaw/Ssal-KRED_evolution.

### Biocatalytic reactions
Small-scale reactions were conducted in 0.2–1.2 ml (Supplementary Method 1.8). Preparative-scale reactions were carried out in 100 ml using iPrOH or glucose as final reductant (Supplementary Method 1.9). After a given time, reactions were quenched with HPLC-grade methanol and analyzed by achiral HPLC-UV/MS (260 nm) or chiral HPLC-UV/MS. An Agilent 1290 HPLC system and corresponding software (Agilent OpenLAB CDS 2.4, Build 2.204.0661) were utilized for data acquisition and analysis (Supplementary Method 1.10 and Supplementary Figs. 4 and 13–17).

### Determination of kinetic constants
Kinetic parameters toward **1a** were measured using purified His-tagged proteins and the UV assay (Supplementary Method 1.11), after confirming that analysis by HPLC-UV (260 nm) at different time points and the UV-based kinetic assay (340 nm) led to comparable values in case of wild-type *Ssal*-KRED (Supplementary Fig. 19). GraphPad Prism (9.2.0) was used for non-linear curve-fitting.

### Substrate binding mode models of *Ssal*-KRED wild type and variants
To construct a binding mode model of **1a**, the NADPH-bound X-ray structure of *Ssal*-KRED (PDB ID: 1Y1P) was used as starting point. First, overlays with other oxidoreductase X-ray structures (PDB ID: 2NNL, 4P38) employing NADPH as redox partner and having the same Ser and Tyr as catalytic residues were generated to identify an anchor for the C=O moiety of **1a**. This information was used to initially place **1a** in the structure 1y1p in an orientation which is consistent with hydride transfer from NADPH to the substrate resulting in the correct *trans*-stereoisomer **2a**. The model was then relaxed using distance constraints of $d$ = 2.8 Å for the interaction of the ligand carbonyl oxygen with Tyr177 OH and Ser133_OG, respectively, as well as $d$ = 3.8 Å for the distance between the carbon atom of NADPH involved in hydride transfer and the ligand carbonyl carbon. Structure superposition was performed using Molecular Operating Environment (MOE 2019.01)[63] and for the model building Moloc[64] was used. Homology models for protein sequences corresponding to hit variants M1, M3 and M6 were built using the same methodology. Each model was visually inspected in PyMOL 2.5.2 and the images were generated using the same software.

### Reporting summary
Further information on research design is available in the Nature Portfolio Reporting Summary linked to this article.

## Data availability
Source data are provided with this paper. Gene and protein sequences are found in Supplementary Information. NMR spectra can be found in

Supplementary Data 1 and Source Data for Fig. 2, Table 2 and Supplementary Figs. 9–11, 19 can be found in Supplementary Data 2–6. The crystal structure used for the substrate docking and homology modeling experiments can be accessed via PDB ID: 1Y1P.

## Code availability

Training data and scripts used to predict enzyme function are available at https://github.com/ccbiozhaw/Ssal-KRED_evolution.

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

## Acknowledgements

The authors thank Robin Küng, Moritz Voss and Takahiro Hayashi for the creation and initial screening of the in-house supplemented KRED collection; Patrik Meier, Veronika Kirrmann and Meinrad Birrer for performing the initial mutational scanning and the preparative experiments to prove the technical relevance of the evolved mutants; Sven Benson and team from Candidum GmbH for their initial support on computer-assisted enzyme design; Fabian Meyer and Peter Stockinger for helpful discussions on enzyme modeling and machine learning, respectively; and Prof. Uwe Bornscheuer for critically reading the manuscript and fruitful feedback. The Zurich University of Applied Sciences and F. Hoffmann-La Roche Ltd. are gratefully acknowledged for financial support. This work was created as part of NCCR Catalysis, a National Centre of Competence in Research funded by the Swiss National Science Foundation (Grant number 180544).

## Author contributions

S.H.M., H.I. and R.B. designed the research and wrote the manuscript. N.D. and S.H.M. set up the lab workflow. J.M., N.D. and S.R. conducted most of the experiments (library construction, HTS, chromatographic analyses, enzyme kinetics and small-scale reactions). M.N. and D.P. implemented bioinformatics tools for data analysis. D.P. carried out the machine learning predictions. S.P.H. and D.W. supervised microbiology and molecular biology at a second site. H.I. set up the screening assay and analytical methods, as well as supervised lab work including scale-up activities at a second site. B.K. performed enzyme modeling and contributed to interpretation. All authors analyzed and discussed the results. H.I. and R.B. led funding acquisition of the collaboration.

## Competing interests

S.P.H., B.K. and H.I. are employees of F. Hoffmann-La Roche Ltd.; D.W. is currently employed at Boehringer-Ingelheim Pharma GmbH & Co KG; J.M. is employed by MSD Werthenstein BioPharma GmbH; and S.R. is employed by Fisher Clinical Services GmbH. The remaining authors declare no competing interests.
