## [Peer Review File · Communications Chemistry]

Reviewers' comments:

Reviewer #1 (Remarks to the Author):

Honda et al., report the directed evolution of a KRED to generate a high-performance enzyme that converts the prochiral ketone 1a into an chiral secondary alcohol (a precursor for Ipatasertib) with high yield, activity, stereoselectivity and substrate loading (100 gL⁻¹). Overall, six rounds of directed evolution introduced ten mutations to generate the final variant M6. The application of the engineered enzyme (M6) on 10g-scale is second highlight of this study. Kinetic analysis revealed that directed evolution generated a biocatalyst with 64-fold increase in k_{cat} and a 12-fold decrease in K_m. Enzyme modelling provided a first rational on the effect of the introduced mutations. Congratulations, these is an extremely impressive study and application of enzyme engineering to yield a useful biocatalyst. The quality of the data is very high, the text is clear and the SI is very informative.

Nevertheless, I'm sorry to say that I can hardly imagine that the manuscript will be published in its current form. In my opinion, the manuscript should not be published as it is. Instead, the manuscript should be rewritten as the authors make very strong claims about the impact of ML on their directed evolution campaign that are not supported by their data and can be very misleading for non-experts (and maybe in part for the field).

To understand my conclusion, it is important to summarize the enzyme engineering efforts: The authors beautifully optimized a wild type KRED in six rounds of directed evolution:

1. Round 1: Six hotspots were identified by mutational scanning. Site-saturation and combinatorial saturation libraries generated the mutant M1 (wild type + F97W / L241M / M242W / Q245S). Factor 8 improvement over the parent.
2. Round 2 introduced two additional mutations that have been previously identified in round 1 to generate M2 (M1+L316M/T342M). Factor 1.1 improvement over the parent. Please note that in this round also an ML-based approach was studied. 30 sequences have been predicted based on a subset of the data from round 1. Yet, this didn't lead to any improvement compared to M1.
3. Round 3 expanded the ML dataset of round 2 to build a larger model (with less experimental coverage of the target sequence space). Almost all the data from round 1 (mainly combinatorial libraries) as well as data from round 2 (30 sequences from the first ML-trial) was included. It is important to understand that this optimized model wasn't used to predict (and test) individual sequences, but rather to identify mutations at target positions for a combinatorial library, which is a smart approach. The authors decided to not explore all six hot spots experimentally, but to focus on three amino acid positions (174, 238 and 241). The corresponding mutations at these positions in the top 50 predictions have been identified with the aim to generate a smarter (smaller) combinatorial library. Until here, this is something that can be described as a ML-filtered combinatorial library. Yet, the authors decided to not screen such a ML-filtered combinatorial library but added (and deleted) mutations to the library that have been found to be beneficial (or non-beneficial) in screening in round 1. So the overall combinatorial library is not purely predicted (as the manuscript text suggests). Instead, the final library contains mutations that are not

predicted by the ML-model and neglects mutations that are predicted by the model. This approach is absolutely OK. But from an engineering perspective, this approach moves away from a ML-filtered combinatorial library to a library which is typically created in enzyme engineering, recombination of beneficial mutations. Screening of this combinatorial library (174A/M/L/V/I, 238L/R/K/G/N, 241M/Q/S) identified the variant M6 (M5+ A238K) with a factor 2.5 improvement over the parent M5. Please note that A238K was also the top mutation in the A238X library (round 1, see Table S5) showing a factor 2.1 increase in activity in round 1. It is true that the final “ML-filtered” combinatorial library also contained ML-predicted mutations that have not been beneficial in round 1, yet, this “novel” mutations didn’t lead to anything better than the A238K mutation identified from screening in round 1. So strong claims about the success of ML on this enzyme engineering campaign are in my opinion not supported by the data.

4. Round 4 introduced a single point mutation to generate M4 (M3+Y246G) by screening a classical combinatorial saturation library of two AA. Factor 1.1 improvement over the parent.

5. Round 5 introduced a single point mutation to generate M5 (M4+ S224A) by screening six SSM libraries. Factor 1.2 improvement over the parent.

6. Round 6 introduced a single point mutation to generate the final variant M6 (M5+ T134V) by screening a combinatorial saturation library of two AA. Factor 2.0 improvement over the parent. Overall ca. factor 60 improvement over the wildtype, in analogy to the increase in kcat, which is great to see.

As partly mentioned above, several additional engineering approaches have been explored during the directed evolution campaign reported here. Yet, they did not lead to improvements (e.g., ML-based prediction in round 2). It is great to see that the authors describe this in detail and explore such potentially very important ML-based enzyme engineering efforts. Yet, at the current stage, this particular directed evolution experiment is in my opinion not driven by ML (see title).

So, what did the ML models contribute to the engineered KRED? To be honest, in my opinion not particular much. It does not mean that the ML-part of the study is not useful. The authors argue that ML significantly reduced the library size in round 3. This is in my interpretation only partly correct, because the combinatorial library of round 3 (75-variants) was only in part predicted by the ML-model and also included beneficial mutations identified from screening in round 1 (even if their ML-model predicted otherwise). The benefit of ML for this round of evolution is interesting, although it is not a purely predicted ML-library. The overall benefit of ML models to the directed evolution of the KRED is in my opinion very limited, yet, the authors make very strong claims about it. I’m sorry to say that, but some statements/conclusions are in my point of view made up and likely very misleading for non-experts. To support publication, the authors should change the following conclusions and statements and more critically reflect on the benefit of ML in this enzyme engineering effort:

- Title: “Machine Learning-Driven Engineering ...” To me, this title suggests that ML had significant impact on enzyme engineering of M6, yet, none of the variants or combinations thereof that are in the final variant have been predicted by the algorithm. It is true that various ML-based approaches have been studied in this study, which is important and great to see. Yet, the only thing one might claim about ML in this enzyme engineering endeavor is the reduction of the library size in round 3 (of overall six

rounds). And this library reduction is only partly based on ML-prediction as the library design in round 3 to a large extent is based on beneficial (non-)behaviour of single points mutations. So one might argue that the library in round 3 is as much a “classical” recombination of beneficial mutations as it is a ML-predicted library (see discussion below). To me, “ML-driven” would mean that a majority of libraries in such a laboratory evolution would originate/predicted by algorithm. I think the authors should use a different title which does not “oversell” the impact of ML on this particular enzyme evolution experiment. The authors put much effort into this project to demonstrate a kind of ML-based engineering concept, yet, a frustrating but maybe fair conclusion is that in the current stage ML did not drive this evolution, if something, it may have made it more expensive. It does not mean that this is not excellent work. A potential title could be “Effective Engineering of a Ketoreductase for the Efficient Synthesis of an Ipatasertib Precursor”? The authors could then mention in the abstract that they have studied ML-based approaches to speed up evolution, and that the impact will be discussed in the manuscript.

- Abstract: The authors write: “Harnessing the power of algorithm-assisted enzyme engineering, we created a 10-amino acid variant exhibiting a 64-fold higher apparent k_{cat} and improved robustness under process conditions compared to the wild-type enzyme”. I believe that many readers will interpret this in the following way: Many of the 10 mutations have been identified based on an algorithm-assisted approach and this ML-method significantly increased k_{cat} . From my point of view this is not true and authors should tone down the importance of ML in their evolution. In my opinion, the following summary much better describes the enzyme engineering used in this study: The authors should highlight that various directed evolution techniques have been harnessed to create the final variant! The abstract should contain information about ML, such as “ML-assisted engineering was partly used to reduce the library size in one round of evolution” or “the benefit of algorithm-assisted enzyme engineering was studied and it was found that ML can be useful to reduce the library size”. The achieved results are fantastic and to me this would be a much better manuscript if the authors would not push so hard to make this manuscript an ML-story.

- Line 207: The authors write: “Instead of ordering distinct enzyme sequences, we constructed a ML-filtered library L10 consisting of 75 variants, in which substrate binding sites L174, A238 and L241 were modulated to a small set of predicted amino acids while M242 and Q245 were fixed to tryptophan and serine, respectively.” The way the combinatorial library design is here presented, suggests that it is completely based on ML-prediction, which is misleading. The authors should mention in the manuscript that part of the predictions have not been considered (based on screening data in earlier rounds) and that the library also contained additional mutations that have not been predicted by ML (based on screening data in earlier rounds). It might even be fair to say that this is as much as recombination library as anything else, or do I completely misunderstand it? Currently, the design of combinatorial library (75 variants, round 3) is hidden in the supplementary text of table S14 and the manuscript generates the impression that this is a ML-predicted library.

- Line 215: The authors write: “Screening of ML-derived library L10 led to the identification of variant M3 (F97W/A238K/L241M/M242W/Q245S/L316M/T342M), which displayed a FLOWT of 22 ... , highlighting the effectiveness of our approach”. This presentation of the results is misleading as it generates the impression that the ML-library was particularly effective (factor 22). Please highlight in the manuscript that

the “ML-filtered” library introduced one mutation into M2 that lead to a 2.5-fold increase compared to the parent, so that readers better understand the success of this particular library. As discussed above, please clearly mention in the manuscript that this library is not a ML-predicted library.

- Discussion: In general, the authors want to highlight the importance of ML for their enzyme engineering. Which is OK. Yet, if ML-assistance in enzyme engineering should really be the story, the readers would appreciate a more scientific discussion about the impact of ML in this enzyme engineering program. Please add a couple of sentences to the discussion which do not only highlight the benefits (reduced library size) but also the discuss the efforts (e.g., sequencing of thousands of mutants) in this ML-based approach used here. The manuscript should contain a scientific discussion / comparison of enzyme engineering efficiencies with both approaches (classical iterative screening versus ML-prediction). As summarized above, this would provide a much clearer picture about the impact and current “burden” of ML-prediction. I really hope that the authors find my comments convincing. To me, such a discussion would be an upgrade to the manuscript and to the fantastic work presented here (as it would help the community avoid the trap of an ML-bubble).

- Conclusion: The authors write: “In our enzyme evolution campaign, we found that an engineering strategy consisting of enzyme hot spot identification through mutational scanning libraries followed by the set-up of structure-guided libraries for machine learning was a particularly effective approach to yield a performant, industrial biocatalyst.” This is misleading as it generates the impression that ML was particularly useful in this enzyme engineering project. Another (and in my opinion fairer) conclusion is that ML efforts have been rather disappointing. I wish the authors would more critically reflect on the success of the ML-based approach compared to the success of classical iterative screening approaches such as site saturation mutagenesis (see round 1, here as mutational scanning as well as round 5), combinatorial libraries (successful in round 1, 4 and 6) or recombination of beneficial mutations from SSM (successful in round 2). The only round in which ML made impact is round 3 and this is not a purely ML-predicted library (see comments above). Frankly, other people might have simply taken variant M2 and recombined the best mutation from the two remaining hot spots (L174A, A238K) into M2 which would have led to the same variant M3. This is of course easy to say afterwards but many enzyme engineers would typically recombine the best mutations identified in previous rounds. Of course, if every single beneficial mutation is included in such a recombination library, even the ones that are only a tiny bit beneficial, then the numbers of variants get very high and difficult to screen (29400 variants, see Fig. 3c). However, from the data identified in L2 (see table S5) some people would have also generated a simply combinatorial library of the most beneficial mutations, such as 97F/W, 174L/Y/A, 238K/Q/R/M/Y, 241K/M/R/W, 242F/W and 245H/N/S/T (overall only 960 variants). Screening such a recombination library (which is not unusual and rather common) would have also led to the same mutations found in M3. This does not mean that the ML-studies by Honda et al. are not particular important. ML-based library design might have a big future, yet, the claims about the impact of ML made by the authors is not supported by their data.

- Figure 2b/c and Table 1: This figure and table suggest that round 3 was based on ML-prediction. To me this is only partly true as the library is also partly based on “classical” recombination of beneficial mutations and in part ignores the ML-predictions. Or do I misunderstand the library design. Example: Position 174 in this combinatorial library included amino acids A,V,I, also they have not been predicted

by ML (please see more examples in Table S14).

Overall, I hope my comments do not sound too critical or even unfriendly. Validation of ML-based approaches to speed up directed evolution campaigns are timely and important. The work by Honda et al. is outstanding enzyme engineering work. I fully support publication if the authors clearly address the comments mentioned above. Please do not hesitate to contact me in case comments are unclear. I hope this great results can be published as soon as possible. From my perspective, no additional experiments are needed.

Minor comments:

Abstract: The abstract mention a 2nd generation, it is not clear what this means. The 1st generation is not mentioned in the manuscript. The authors may want to delete this from the abstract or add an explanation to the manuscript.

Line 32: The authors may want to put this information in two independent sentences. The sentence starting with "Furthermore (line 31) is a bit too long.

Line 82: The text says 0.3% (w/v) substrate loading, The figure caption (Fig. 1) says 10 mM substrate loading. The authors may want to provide g/L substrate loading in the whole manuscript, to make it easier for the reader (100 g/L is mentioned earlier as aim in the manuscript).

Line 291: Please define s/e ratio. Can the authors please also provide mol% KRED that have been applied, so that readers better understand the importance of the data. Many readers might not be familiar with the term s/e ratio and they might also not be able to make an interpretation of this value.

Line 311: Please change to "12-fold lower" as the K_m decreased from wild type to M6. Or do I misunderstand something?

Line 398: Do the authors mean a "high-performance, industrial biocatalyst"?

SI, Fig S3: It is not clear what t-SNE means. Can the authors please describe what this is and how it was generated?

Reviewer #2 (Remarks to the Author):

In this manuscript, the authors engineered a ketoreductase to the extent that the enzyme could be used in industry, by using a variety of molecular evolution techniques. Catalytic activity of more than 50 enzyme candidates was experimentally measured to select a parent enzyme, and the molecular evolutions of point/combinatorial saturation mutagenesis were iteratively applied. In this process,

machine learning was also applied to identify the apparent amino acids in some sites and the utilization of machine learning was contributed to decrease sequence space. Finally, the engineered enzyme showed 64-fold higher apparent k_{cat} . Their results are clear and this is an interesting manuscript in which the function of enzyme was maximized by combinatorial protein engineering. I can recommend this manuscript to the publication on Communications Chemistry, but I feel that the contribution of machine learning.

1.Title: I feel that the contribution of machine learning. The authors should remove the word of machine learning.

2.Library of L3-6: In each library, 3 sites were selected from 5 sites. I think that the number of combinations is 10 ($5C3$). They should explain why the combinations of 3 sites in L3, L4, L5, L6 was selected to experimentally analyze the activity.

3.Library of L8: In this library, four sites (144, 150, 316, 326, 342) were selected for mutagenesis. The sequence space is 20^5 , but 31 variants were experimentally analyzed. Why the 31 variants were selected to measure the activity ? The authors should explain it in detail.

Reviewer #3 (Remarks to the Author):

Recommendation: Suitable for publication in Communication Chemistry after revision.

The paper discusses the application of machine learning in the engineering campaign of a ketoreductase from *Sporidiobolus salmonicolor* for the synthesis of ipatasertib, a protein kinase B inhibitor. The engineered enzyme variant showed improvements in apparent k_{cat} and robustness under process conditions compared to the wild-type enzyme. The paper addresses a relevant issue in enzyme engineering, specifically focusing on the enzyme activity for the synthesis of a biologically active compound. The successful implementation of the engineered enzyme in the chemo-enzymatic synthesis of ipatasertib, with high conversion rates and diastereomeric excess, highlights the practical applicability of the proposed methodology. The paper is generally well-written and presents a valuable contribution to the field of enzyme engineering. However, one concern I have with the manuscript is the depth of presentation of ML results. The work may benefit from a deeper explanation of choice of ML methodology, the benchmark results, validation and deployment of models and further discussion where ML-informed DE has been particularly successful. Authors generated a valuable resource of mutational data, they can probably analysis and benefit more and reflect on non-linearity effects, i.e., non-additive impacts, resulting from interactions between mutations within a protein sequence. As far as I am aware the proposed strategy is not fully a new approach for enzyme evolution, but a successful case study is demonstrated. This suggests potential for the future. Thus, I believe the paper deserves publication in Communication Chemistry after addressing these comments in preparing the revision. The comments are from the point of view of a reader deeper in ML guided enzyme engineering. I believe that addressing the mentioned points could further improve the clarity and completeness of the manuscript. Hopefully other reviewers will have more technical expertise in the biocatalysis and

molecular biology part.

Specific comments:

1. The abstract could benefit from providing more details on the specific machine learning algorithms employed in the enzyme engineering process. A brief mention of the algorithm types used, and their roles could enhance the clarity of the methodology.
2. It would be valuable to discuss how this approach compares with other existing methods for enzyme engineering, especially regarding the improvement of enzyme activity for challenging substrates.
3. The authors have provided several intriguing insights. However, I believe there is insufficient bibliographic coverage on several points (regarding the ML methodology), which is necessary to justify the application. Additionally, the paper discusses the capabilities of ML methods, but it lacks detailed information on the specific ML techniques that were used (apart from Gaussian processes). Provided supplementary github link (not available) and materials does not provide much details on the ML models and training.
4. The training set is the large number of single point and combinatorial mutations. New mutations are predicted and tested by rigorous measurements, which are well-performed. There are some successes in the approach, with predicted multiple mutation proteins having better performance, as confirmed by experiment. Author might discuss problems and uncertainties associated with the sparse and uneven training set, the results are nonetheless impressive, and the effect of the predicted mutations could be discussed further in terms of known effects on enzyme activity or non-linear interactions.
5. Some discussion on the inclusion and impact of negative variants on ML prediction of combinatorial variants and mutational trajectories would be very helpful.

While the abstract mentions modeling and kinetic studies, providing a bit more information on the key findings from these studies could enhance the understanding of the mechanistic factors governing the improved reaction outcome.

Some minor typos or corrections include:

-Strictly speaking paper sometimes suffers from unclear language or imprecise statements. For instance, these sentences in the abstract have to be:

“While the improvement of protein structure, stability and solubility has been frequently explored in this way, enzyme activity has proven to be a more difficult target function.” \diamond it means improvement of prediction of protein structure, stability and solubility ... \diamond as we can not improve the protein structure rather its prediction, sentence is correct for the stability and solubility.

Additionally, a comprehensive review in this field (<https://doi.org/10.1021/acscatal.3c02743>) shows several examples of ML- guided engineering of enzyme activity. Which contradicts the statement made by authors that “enzyme activity has proven to be a more difficult target function”.

“Harnessing the power of algorithm-assisted enzyme engineering, we created a 10-amino acid variant exhibiting a 64-fold higher apparent k_{cat} and improved robustness under process conditions compared to the wild-type enzyme.” \diamond 10-amino acid variant is misleading \diamond it should be described as e.g. 10-amino acid substituted variant or a variant harboring 10 substitutions.

-Line 65, references 3233 should be either comma or dash separated.

Reviewer #1 (Remarks to the Author):

Honda et al., report the directed evolution of a KRED to generate a high-performance enzyme that converts the prochiral ketone 1a into a chiral secondary alcohol (a precursor for lpatasertib) with high yield, activity, stereoselectivity and substrate loading (100 gL⁻¹). Overall, six rounds of directed evolution introduced ten mutations to generate the final variant M6. The application of the engineered enzyme (M6) on 10g-scale is second highlight of this study. Kinetic analysis revealed that directed evolution generated a biocatalyst with 64-fold increase in kcat and a 12-fold decrease in Km. Enzyme modelling provided a first rational on the effect of the introduced mutations. Congratulations, these is an extremely impressive study and application of enzyme engineering to yield a useful biocatalyst. The quality of the data is very high, the text is clear and the SI is very informative.

We are grateful to Reviewer 1 for the careful reviewing our manuscript and his/her valuable feedback on how to improve it.

Nevertheless, I'm sorry to say that I can hardly imagine that the manuscript will be published in its current form. In my opinion, the manuscript should not be published as it is. Instead, the manuscript should be rewritten as the authors make very strong claims about the impact of ML on their directed evolution campaign that are not supported by their data and can be very misleading for non-experts (and maybe in part for the field).

To understand my conclusion, it is important to summarize the enzyme engineering efforts: The authors beautifully optimized a wild type KRED in six rounds of directed evolution:

1. Round 1: Six hotspots were identified by mutational scanning. Site-saturation and combinatorial saturation libraries generated the mutant M1 (wild type + F97W / L241M / M242W / Q245S). Factor 8 improvement over the parent.

2. Round 2 introduced two additional mutations that have been previously identified in round 1 to generate M2 (M1+L316M/T342M). Factor 1.1 improvement over the parent. Please note that in this round also an ML-based approach was studied. 30 sequences have been predicted based on a subset of the data from round 1. Yet, this didn't lead to any improvement compared to M1.

3. Round 3 expanded the ML dataset of round 2 to build a larger model (with less experimental coverage of the target sequence space). Almost all the data from round 1 (mainly combinatorial libraries) as well as data from round 2 (30 sequences from the first ML-trial) was included. It is important to understand that this optimized model wasn't used to predict (and test) individual sequences, but rather to identify mutations at target positions for a combinatorial library, which is a smart approach. The authors decided to not explore all six hot spots experimentally, but to focus on three amino acid positions (174, 238 and 241). The corresponding mutations at these positions in the top 50 predictions have been identified with the aim to generate a smarter (smaller) combinatorial library. Until here, this is something that can be described as a ML-filtered combinatorial library. Yet, the authors decided to not screen such a ML-filtered combinatorial library but added (and deleted) mutations to the library that have been found to be beneficial (or non-beneficial) in screening in round 1. So the overall combinatorial library is not purely predicted (as the manuscript text suggests). Instead, the final library contains mutations that are not predicted by the ML-model and neglects mutations that are predicted by the model. This approach is absolutely OK. But from an engineering perspective, this approach moves away from a ML-filtered combinatorial library to a library which is typically created in enzyme engineering, recombination of beneficial mutations. Screening of this combinatorial library (174A/M/L/V/I, 238L/R/K/G/N, 241M/Q/S) identified the variant M6 (M5+ A238K) with a factor 2.5 improvement over the parent M5. Please note that A238K was also the top mutation in the A238X library (round 1, see Table S5) showing a factor 2.1 increase in activity in round 1. It is true that the final "ML-filtered" combinatorial library also contained ML-predicted mutations that have not been beneficial in round 1, yet, this "novel" mutations didn't lead to anything better than the A238K mutation identified from screening in round 1. So strong claims about the success of ML on this enzyme engineering campaign are in my opinion not supported by the data.

4. Round 4 introduced a single point mutation to generate M4 (M3+Y246G) by screening a classical combinatorial saturation library of two AA. Factor 1.1 improvement over the parent.

5. Round 5 introduced a single point mutation to generate M5 (M4+ S224A) by screening six SSM libraries. Factor 1.2 improvement over the parent.

6. Round 6 introduced a single point mutation to generate the final variant M6 (M5+ T134V) by screening a combinatorial saturation library of two AA. Factor 2.0 improvement over the parent. Overall ca. factor 60 improvement over the wildtype, in analogy to the increase in kcat, which is great to see.

As partly mentioned above, several additional engineering approaches have been explored during the directed evolution campaign reported here. Yet, they did not lead to improvements (e.g., ML-based prediction in round 2). It is great to see that the authors describe this in detail and explore such potentially very important ML-based enzyme engineering efforts. Yet, at the current stage, this particular directed evolution experiment is in my opinion not driven by ML (see title).

So, what did the ML models contribute to the engineered KRED? To be honest, in my opinion not particular much. It does not mean that the ML-part of the study is not useful. The authors argue that ML significantly reduced the library size in round 3. This is in my interpretation only partly correct, because the combinatorial library of round 3 (75-variants) was only in part predicted by the ML-model and also included beneficial mutations identified from screening in round 1 (even if their ML-model predicted otherwise). The benefit of ML for this round of evolution is interesting, although it is not a purely predicted ML-library. The overall benefit of ML models to the directed evolution of the KRED is in my opinion very limited, yet, the authors make very strong claims about it. I'm sorry to say that, but some statements/conclusions are in my point of view made up and likely very misleading for non-experts. To support publication, the authors should change the following conclusions and statements and more critically reflect on the benefit of ML in this enzyme engineering effort:

We thank Reviewer 1 for his/her careful analysis of our manuscript. We agree that our semi-rational enzyme evolution strategy played a very important part in boosting enzymatic activity. Following Reviewer 1's concrete suggestions below, we have therefore rewritten the manuscript to give a more balanced overview of the contributions of the individual enzyme variant libraries.

- Title: "Machine Learning-Driven Engineering ..." To me, this title suggests that ML had significant impact on enzyme engineering of M6, yet, none of the variants or combinations thereof that are in the final variant have been predicted by the algorithm. It is true that various ML-based approaches have been studied in this study, which is important and great to see. Yet, the only thing one might claim about ML in this enzyme engineering endeavor is the reduction of the library size in round 3 (of overall six rounds). And this library reduction is only partly based on ML-prediction as the library design in round 3 to a large extent is based on beneficial (non-)behaviour of single points mutations. So one might argue that the library in round 3 is as much a "classical" recombination of beneficial mutations as it is a ML-predicted library (see discussion below). To me, "ML-driven" would mean that a majority of libraries in such a laboratory evolution would originate/predicted by algorithm. I think the authors should use a different title which does not "oversell" the impact of ML on this particular enzyme evolution experiment. The authors put much effort into this project to demonstrate a kind of ML-based engineering concept, yet, a frustrating but maybe fair conclusion is that in the current stage ML did not drive this evolution, if something, it may have made it more expensive. It does not mean that this is not excellent work. A potential title could be "Effective Engineering of a Ketoreductase for the Efficient Synthesis of an Ipatasertib Precursor"? The authors could then mention in the abstract that they have studied ML-based approaches to speed up evolution, and that the impact will be discussed in the manuscript.

Title was changed to "Effective Engineering of a Ketoreductase for the Biocatalytic Synthesis of an Ipatasertib Precursor", as suggested.

- Abstract: The authors write: "Harnessing the power of algorithm-assisted enzyme engineering, we created a 10-amino acid variant exhibiting a 64-fold higher apparent kcat and improved robustness under process conditions compared to the wild-type enzyme". I believe that many readers will interpret this in the following way: Many of the 10 mutations have been identified based on an algorithm-assisted approach and this ML-method significantly increased kcat. From my point of view this is not true and authors should tone down the importance of ML in their evolution. In my opinion, the following summary much better describes the enzyme engineering used in this study: The authors should highlight that various directed evolution techniques have been harnessed to create the final variant! The abstract should contain information about ML, such as "ML-assisted engineering was partly used to reduce the library size in one round of evolution" or "the benefit of algorithm-assisted enzyme engineering was studied and it was found that ML can be useful to reduce the library size". The achieved results are fantastic and to me this would be a much better manuscript if the authors would not push so hard to make this manuscript an ML-story.

The abstract was rewritten according to the reviewer's suggestions.

- Line 207: The authors write: "Instead of ordering distinct enzyme sequences, we constructed a ML-filtered library L10 consisting of 75 variants, in which substrate binding sites L174, A238 and L241 were modulated to a small set of predicted amino acids while M242 and Q245 were fixed to tryptophan and serine, respectively." The way the combinatorial library design is here presented, suggests that it is completely based on ML-prediction, which is misleading. The authors should mention in the manuscript that part of the predictions have not been considered (based on screening data in earlier rounds) and that the library also contained additional mutations that have not been predicted by ML (based on screening data in earlier rounds). It might even be fair to say that this is as much as recombination library as anything else, or do I completely misunderstand it? Currently, the design of combinatorial library (75 variants, round 3) is hidden in the supplementary text of table S14 and the manuscript generates the impression that this is a ML-predicted library.

We apologize if our description of the set-up of the ML library was not clear enough in the main text and SI. To illustrate that we used additional data to construct the library, we had employed the term "filtered" ML library. Following reviewer 1's suggestion, we now give a more detailed description of library design to avoid any ambiguities.

- Line 215: The authors write: "Screening of ML-derived library L10 led to the identification of variant M3 (F97W/A238K/L241M/M242W/Q245S/L316M/T342M), which displayed a FLOWT of 22 ... , highlighting the effectiveness of our approach". This presentation of the results is misleading as it generates the impression that the ML-library was particularly effective (factor 22). Please highlight in the manuscript that the "ML-filtered" library introduced one mutation into M2 that led to a 2.5-fold increase compared to the parent, so that readers better understand the success of this particular library. As discussed above, please clearly mention in the manuscript that this library is not a ML-predicted library.

We have rewritten the sentence accordingly.

- Discussion: In general, the authors want to highlight the importance of ML for their enzyme engineering. Which is OK. Yet, if ML-assistance in enzyme engineering should really be the story, the readers would appreciate a more scientific discussion about the impact of ML in this enzyme engineering program. Please add a couple of sentences to the discussion which do not only highlight the benefits (reduced library size) but also discuss the efforts (e.g., sequencing of thousands of mutants) in this ML-based approach used here. The manuscript should contain a scientific discussion / comparison of enzyme engineering efficiencies with both approaches (classical iterative screening versus ML-prediction). As summarized above, this would provide a much clearer picture about the impact and current "burden" of ML-prediction. I really hope that the authors find my comments convincing. To me, such a discussion would be an upgrade to the manuscript and to the fantastic work presented here (as it would help the community avoid the trap of an ML-bubble).

We have rephrased the discussion of our results, as suggested by reviewer 1, and clearly pointed out that the machine learning was but one aspect of the entire evolution campaign.

- Conclusion: The authors write: "In our enzyme evolution campaign, we found that an engineering strategy consisting of enzyme hot spot identification through mutational scanning libraries followed by the set-up of structure-guided libraries for machine learning was a particularly effective approach to yield a performant, industrial biocatalyst." This is misleading as it generates the impression that ML was particularly useful in this enzyme engineering project. Another (and in my opinion fairer) conclusion is that ML efforts have been rather disappointing. I wish the authors would more critically reflect on the success of the ML-based approach compared to the success of classical iterative screening approaches such as site saturation mutagenesis (see round 1, here as mutational scanning as well as round 5), combinatorial libraries (successful in round 1, 4 and 6) or recombination of beneficial mutations from SSM (successful in round 2). The only round in which ML made impact is round 3 and this is not a purely ML-predicted library (see comments above). Frankly, other people might have simply taken variant M2 and recombined the best mutation from the two remaining hot spots (L174A, A238K) into M2 which would have led to the same variant M3. This is of course easy to say afterwards but many enzyme engineers would typically recombine the best mutations identified in previous rounds. Of course, if every single beneficial mutation is included in such a recombination library, even the ones that are only a tiny bit beneficial, then the numbers of variants get very high and difficult to screen (29400 variants, see Fig. 3c). However, from the data identified in L2 (see table S5) some people would have also generated a simply combinatorial library of the most beneficial mutations, such as 97F/W, 174L/Y/A, 238K/Q/R/M/Y, 241K/M/R/W, 242F/W and 245H/N/S/T (overall only 960 variants). Screening such a recombination library (which is not unusual and rather common) would have also led to the same mutations found in M3. This does not mean that the ML-studies

by Honda et al. are not particularly important. ML-based library design might have a big future, yet, the claims about the impact of ML made by the authors is not supported by their data.

We would like to point out that the screening of the 5-site CSM library was rather disappointing, leaving us with a FLOWT of only 3.4 (compared to a FLOWT of 3.6 from the best single mutant). As a result, we were very happy to find that the filtered ML library was successful to boost us by an additional factor of 2.5.

We agree with reviewer 1 that it is always possible to choose alternative ways to combine beneficial mutations from previous rounds of evolution. However, except if screening all possible variants (typically ensured by oversampling), it is very difficult to make sure that the “best possible” variant is being identified in the given design space. Retrospectively, we agree, that combining the most beneficial substitutions in the SSM libraries could have led to the identification of improved variant, maybe even M6. However, at this point of the evolution, we opted to apply the filtered ML approach, e.g., using predictions and combining them with prior knowledge. Interestingly, this approach allowed us to identify several amino acid substitutions in other improved variants (such as A238G or A238L), which were neutral or deleterious in the context of the wildtype enzyme.

We would like to highlight again that the new version of the manuscript strives to give a balanced overview of the overall evolution trajectory.

- Figure 2b/c and Table 1: This figure and table suggest that round 3 was based on ML-prediction. To me this is only partly true as the library is also partly based on “classical” recombination of beneficial mutations and in part ignores the ML-predictions. Or do I misunderstand the library design. Example: Position 174 in this combinatorial library included amino acids A,V,I, also they have not been predicted by ML (please see more examples in Table S14).

To explain our L10 library design in more detail (filtered ML library), we expanded the main text as well as the figure legends with information about how we combined the ML predictions with prior experimental data to derive the final library composition. Furthermore, we relabeled the library description in Table 1 from “ML” to “filtered ML”.

Overall, I hope my comments do not sound too critical or even unfriendly. Validation of ML-based approaches to speed up directed evolution campaigns are timely and important. The work by Honda et al. is outstanding enzyme engineering work. I fully support publication if the authors clearly address the comments mentioned above. Please do not hesitate to contact me in case comments are unclear. I hope this great results can be published as soon as possible. From my perspective, no additional experiments are needed.

Again, we would like to thank reviewer 1 for the thoughtful feedback. We are convinced that the manuscript profited from the changes made.

Minor comments:

Abstract: The abstract mention a 2nd generation, it is not clear what this means. The 1st generation is not mentioned in the manuscript. The authors may want to delete this from the abstract or add an explanation to the manuscript.

For clarity, we deleted the term “2nd generation” from the abstract. Nevertheless, we would like to point out, that we do mention the 1st generation process in the introduction.

Line 74: “One of these chiral centers is currently introduced by a commercial KRED which is capable of asymmetrically reducing the prochiral ketone **1a** to the desired (R,R)-trans alcohol intermediate **2a** (Scheme 1).”

Line 32: The authors may want to put this information in two independent sentences. The sentence starting with “Furthermore (line 31) is a bit too long.

We have rewritten the sentence in the following way:

“Notably, recycling of the cofactor on industrial scale has been well established using isopropanol as cheap hydride donor.¹⁰ In addition, the enzymes are produced from renewable resources and are readily biodegradable enabling sustainable industrial processes.”

Line 82: The text says 0.3% (w/v) substrate loading, The figure caption (Fig. 1) says 10 mM substrate loading. The authors may want to provide g/L substrate loading in the whole manuscript, to make it easier for the reader (100 g/L is mentioned earlier as aim in the manuscript).

We have standardized the units to g L⁻¹ throughout the manuscript and the SI. In Fig. 1 both g L⁻¹ and mM units are indicated since the *cis*- and *trans*-alcohol products from the KRED collection screening were quantified in mM.

Line 291: Please define s/e ratio. Can the authors please also provide mol% KRED that have been applied, so that readers better understand the importance of the data. Many readers might not be familiar with the term s/e ratio and they might also not be able to make an interpretation of this value.

A description for s/e has been added to the text in Page 8 and in Table 2. However, we are unable to provide mol% KRED as the enzymes are not applied in purified form, but rather lyophilized enzyme lysates were used. For clarity, we have added the g of KRED (as lyo enzyme lysate) (Table 2).

Line 311: Please change to “12-fold lower” as the K_m decreased from wild type to M6. Or do I misunderstand something?

Current phrasing, e.g., “12-fold higher” is correct.

During evolution, we increased both k_{cat} and K_m . While the k_{cat} increased from 0.49 min⁻¹ (wildtype) to 31.5 min⁻¹ (M6), the K_m increased from 11.9 μ M (wildtype) to 143.5 μ M (M6), presumably because there was no evolutionary pressure for higher binding affinity to the substrate (substrate load was always very high in the screening assays).

Line 398: Do the authors mean a “high-performance, industrial biocatalyst”?

Corrected accordingly.

SI, Fig S3: It is not clear what t-SNE means. Can the authors please describe what this is and how it was generated?

t-SNE refers to “t-distributed Stochastic Neighbor Embedding”, an unsupervised statistical non-linear dimensionality reduction method for data exploration and visualization of high-dimensional data. It converts similarities between data points to joint probabilities and is used to understand underlying patterns and relationships in the data (L. van der Maaten, *J Machine Learning* 2008, 9, 2579 - 2605). The method’s full name and the corresponding citation have been added to the figure legend and to the list of references in the SI, respectively.

Reviewer #2 (Remarks to the Author):

In this manuscript, the authors engineered a ketoreductase to the extent that the enzyme could be used in industry, by using a variety of molecular evolution techniques. Catalytic activity of more than 50 enzyme candidates was experimentally measured to select a parent enzyme, and the molecular evolutions of point/combinatorial saturation mutagenesis were iteratively applied. In this process, machine learning was also applied to identify the apparent amino acids in some sites and the utilization of machine learning was contributed to decrease sequence space. Finally, the engineered enzyme showed 64-fold higher apparent k_{cat} . Their results are clear and this is an interesting manuscript in which the function of enzyme was maximized by combinatorial protein engineering. I can recommend this manuscript to the publication on CommunicationsChemistry, but I feel that the contribution of machine learning.

We are grateful to Reviewer 1 for the careful reviewing our manuscript and his/her valuable feedback on how to improve it.

1.Title: I feel that the contribution of machine learning. The authors should remove the word of machine learning.

Title was changed to “Effective Engineering of a Ketoreductase for the Biocatalytic Synthesis of an Ipatasertib Precursor”, as suggested.

2. Library of L3-6: In each library, 3 sites were selected from 5 sites. I think that the number of combinations is 10 (5C3). They should explain why the combinations of 3 sites in L3, L4, L5, L6 was selected to experimentally analyze the activity.

Due to limited resources, we had to reduce the number of 3-site combinatorial libraries to be screened. As these libraries were intended for further processing via machine learning, we opted to select combinations based on geometrical considerations, combining mutations to be most likely to interact. To make this clearer to the reader, we extended our reasoning in the main text and refer to Figure 3b.

“Due to time- and resource restriction, we opted to limit our efforts to four out of the possible ten 3-site combinatorial libraries. The library design was guided by geometrical considerations leading us to group amino acid positions most likely to interact (Figure 3b).”

3. Library of L8: In this library, four sites (144, 150, 316, 326, 342) were selected for mutagenesis. The sequence space is 20^5 , but 31 variants were experimentally analyzed. Why the 31 variants were selected to measure the activity? The authors should explain it in detail.

These positions were not subjected to saturation mutagenesis, but rather substituted with only two amino acids: the native amino acid and the beneficial mutation found in the mutational scanning library L1. Therefore, the sequence space = $2^5 = 32$ variants – 1 (WT which originates from the combination of only the native residue on each position) = 31. For clarity, this explanation has been introduced in the footnote of Table S8.

Reviewer #3 (Remarks to the Author):

Recommendation: Suitable for publication in Communication Chemistry after revision.

The paper discusses the application of machine learning in the engineering campaign of a ketoreductase from *Sporidiobolus salmonicolor* for the synthesis of ipatasertib, a protein kinase B inhibitor. The engineered enzyme variant showed improvements in apparent k_{cat} and robustness under process conditions compared to the wild-type enzyme. The paper addresses a relevant issue in enzyme engineering, specifically focusing on the enzyme activity for the synthesis of a biologically active compound. The successful implementation of the engineered enzyme in the chemo-enzymatic synthesis of ipatasertib, with high conversion rates and diastereomeric excess, highlights the practical applicability of the proposed methodology. The paper is generally well-written and presents a valuable contribution to the field of enzyme engineering. However, one concern I have with the manuscript is the depth of presentation of ML results. The work may benefit from a deeper explanation of choice of ML methodology, the benchmark results, validation and deployment of models and further discussion where ML-informed DE has been particularly successful. Authors generated a valuable resource of mutational data, they can probably analysis and benefit more and reflect on non-linearity effects, i.e., non-additive impacts, resulting from interactions between mutations within a protein sequence. As far as I am aware the proposed strategy is not fully a new approach for enzyme evolution, but a successful case study is demonstrated. This suggests potential for the future. Thus, I believe the paper deserves publication in Communication Chemistry after addressing these comments in preparing the revision. The comments are from the point of view of a reader deeper in ML guided enzyme engineering. I believe that addressing the mentioned points could further improve the clarity and completeness of the manuscript. Hopefully other reviewers will have more technical expertise in the biocatalysis and molecular biology part.

We are grateful to Reviewer #3 for his/her valuable contribution towards enhancing the quality of our manuscript’s ML section, as well as his/her detailed observations regarding typos and minor corrections. Please find below our answers to his/her specific comments and suggestions.

Specific comments:

1. The abstract could benefit from providing more details on the specific machine learning algorithms employed in the enzyme engineering process. A brief mention of the algorithm types used, and their roles could enhance the clarity of the methodology.

We agree and have modified the Abstract as follows (also following suggestions from Reviewer 1):

Semi-rational enzyme engineering is a powerful method to develop industrial biocatalyst. Profiting from advances in molecular biology and bioinformatics, semi-rational approaches can effectively accelerate enzyme engineering campaigns. Here, we present the optimization of a ketoreductase from *Sporidiobolus salmonicolor* for the chemo-enzymatic synthesis of ipatasertib, a potent protein kinase B inhibitor. Harnessing the power of mutational scanning and structure-guided rational design, we created a 10-amino acid substituted variant exhibiting a 64-fold higher apparent k_{cat} and improved robustness under process conditions compared to the wild-type enzyme. In addition, the benefit of algorithm-aided enzyme engineering was studied to derive correlations in protein sequence-function data and it was found that the applied Gaussian processes allowed us to reduce enzyme library size. The final scalable and high performing biocatalytic process yielded the alcohol intermediate with $\geq 98\%$ conversion and a diastereomeric excess of 99.7% (*R,R-trans*) from 100 g L⁻¹ ketone after 30 h. Modelling and kinetic studies shed light on the mechanistic factors governing the improved reaction outcome, with mutations T134V, A238K, M242W and Q245S exerting the most beneficial effect on reduction activity towards the target ketone.

2. It would be valuable to discuss how this approach compares with other existing methods for enzyme engineering, especially regarding the improvement of enzyme activity for challenging substrates.

The ML method applied in this study does not consider substrate information and exclusively relies on protein sequence space. Nonetheless, we have formerly showcased the efficacy of sequence-based Gaussian Processes for Machine Learning (GPML) for improving the activity as well as the chemo- and regio-selectivity of an alpha-ketoglutarate/Fe-dependent halogenase toward non-natural substrates like soraphens. Noteworthy, soraphens are complex and bulky polyketides that significantly differ from indol alkaloids, the native substrates (J. Büchler et al., Nat Commun 2022, 13, 1 – 11 - this paper has been cited). In addition, we concur with the suggestion to compare the ML method used with alternative ones, and have thus incorporated the following discussion in the manuscript:

Next to Gaussian processes, various other ML algorithms have been employed to predict improved protein solubility, stability, specificity, or activity. These methodologies, including partial least-squares regression, random forest, decision trees, support vector machines, *K*-nearest neighbors, and neural networks, have been scrutinized in comprehensive reviews (S. Mazurenko et al., ACS Catal 2020, 10, 2, 1210 – 1223; B. J. Wittmann et al., Curr Opin Struct Biol 2021, 69, 11 – 18; M. Braun et al., ACS Catal 2023, 13, 21, 14454 – 14469; D. Patsch and R. Buller, Chimia 2023, 77, 3, doi:10.2533/chimia.2023.116, B. Hie et al., Cell Syst 2020, 11, 461– 477; B. L. Hie and K. K. Yang, Curr Opin Struct Biol 2022, 72, 145–152).

3. The authors have provided several intriguing insights. However, I believe there is insufficient bibliographic coverage on several points (regarding the ML methodology), which is necessary to justify the application.

*Our ML methodology is based on a previous research paper from our group (J. Büchler et al., Nat Commun 2022, 13, 371, 1 - 11) and we refer to this publication in the Methods section (Manuscript, Page 11). The differences between the methodologies used in the former paper and the present study were briefly explained. Nevertheless, for the sake of clarity, we have added a note leading to Methods in the Results and Discussion section (Manuscript, Page 5). In addition, we have rewritten the Methods section in a more detailed manner (**added information is highlighted in bold**):*

Results and Discussion (Manuscript, Page 5):

Going forward, we set out to explore the remaining protein landscape in silico using Gaussian processes. In analogy to a previous successful application of machine-learning for enzyme design in our laboratory,³⁶ we employed a strategy in which we represented amino acids by their different physicochemical and biochemical characteristics, derived from the AAindex database^{37,38} (**see Methods section**).

Methods (Manuscript, Page 11):

Machine learning. Amino acid sequence and activity data (FIOP values) analyses of library variants were automated (Supplementary Method 1.7) and used as input for machine learning (ML) which was adapted from Buechler *et al.*³⁶ The target label (activity) was calculated by dividing **the slope of** each sample on a plate by the averaged wild-type slope for that plate. The individual variants were represented based on the amino acids' various physicochemical and biochemical properties at the mutated sites. These properties were derived from the the AAindex database,^{37,38} an extensive collection of amino acid characteristics from several sources. **Thirteen features per site, i.e., 13 × 3 (241-242-245 for L9) and 13 × 5 (174-248-241-242-245 for L10), were considered. We then defined the feature vector of a sequence by joining the vector representation of its individual amino acids at the defined sites and aggregated them into a 651 × 39- and a 2453 × 65 -dimensional training matrix for L9 and L10, respectively.** All predictions were made based on the Algorithm 2.1 of

Gaussian Processes for ML (GPML) by Rasmussen and Williams,⁵⁵ implemented in the scikit-learn Python module. **To mitigate overfitting and enhance the assessment of the generalizability of our model, we cross-validated over ten splits, and model performance was evaluated using the coefficient of determination (R²). As a result, average R² scores of 0.66 and 0.77 were achieved for L9 and L10, respectively (compare predicted vs. measured, Figure S4).** The code, data, and related supplementary information can be accessed at https://github.com/ccbiozhaw/Ssal-KRED_evolution.

Additionally, the paper discusses the capabilities of ML methods, but it lacks detailed information on the specific ML techniques that were used (apart from Gaussian processes).

This is related to Comment No. 2/3 and we have addressed this request by adding a paragraph to the discussion section in the manuscript as well as extending the method section to give detailed information about the Gaussian process used (which was the only ML- technique applied in this paper).

Provided supplementary github link (not available) and materials does not provide much details on the ML models and training.

We apologize that the github link given in the manuscript was not available for the reviewers. We did not open the link for public access, but instead provided a ZIP file entitled "Ssal-KRED_evolution-master.zip" upon submission. This folder contains all relevant documents related to the ML section, including a "README.md" file with the description of each file as well as references (please find below). The github link will be opened for public access upon manuscript acceptance.

README.md file:

```
# Paper data

### Data and code to reproduce our results

library 9 and library 10 contain the training data and results for their respective library

library9/10_training.ipynb - jupyter notebook with our code. the cells can be executed in order from top to bottom. The notebook was already pre-run, and the expected outputs are visible. The only requirements are for the data files to be in the same directory as the jupyter notebook file. The runtime for library9 should not exceed a couple of minutes, even on a standard laptop. This increases manyfold for library10, as making the predictions for all 20\*\*5 variants is very expensive. You can reduce this by setting testing to True. To make predictions on your own data, your inputs have to adhere strictly to the format of our own data, or the code has to be adjusted respectively.

library9/10_data.xlsx - the target value is defined as slope variant / slope wild-type. the slope is derived from the UV-assay.

aaindex.csv - AAindex is a database of numerical indices representing various physicochemical and biochemical properties of amino acids and pairs of amino acids. This is used to represent amino acids. [1,2]

[1] Kawashima, S. and Kanehisa, M.; AAindex: amino acid index database. Nucleic Acids Res. 28, 374 (2000). [PMID:10592278]

[2] Kawashima, S., Pokarowski, P., Pokarowska, M., Kolinski, A., Katayama, T., and Kanehisa, M.; AAindex: amino acid index database, progress report 2008. Nucleic Acids Res. 36, D202-D205 (2008).[PMID:17998252]

[...]
```

4. The training set is the large number of single point and combinatorial mutations. New mutations are predicted and tested by rigorous measurements, which are well-performed. There are some successes in the approach, with predicted multiple mutation proteins having better performance, as confirmed by experiment. Author might discuss problems and uncertainties associated with the sparse and uneven training set, the results are nonetheless impressive, and the effect of the predicted mutations could be discussed further in terms of known effects on enzyme activity or non-linear interactions.

Indeed. A concern that arises from the sparse and uneven training dataset is that the sequence space might not be fully explored, and areas of significant improvement are missed. This could be addressed by an iterative approach of balancing exploration and exploitation through Bayesian learning techniques as outlined elsewhere (P. A. Romero et al., *PNAS* **2012**, 110, 3, E193). Therefore, we have modified the manuscript as follows (Results and Discussion, Manuscript, Page 6):

Clearly, the effectiveness of the approach was influenced by the decision to construct an enzyme library, which was informed by machine learning predictions and knowledge about previous successful variants, instead of solely testing a ranked set of predicted sequences. In this context, potential uncertainties of the model were compensated by maximizing the probability via greedy exploitation of promising predicted mutations. An alternative approach to address potential bias introduced by limited and imbalanced training datasets would have been to follow an iterative approach of balancing exploration and exploitation through Bayesian learning techniques (P. A. Romero et al., *PNAS* 2012, 110, 3, E193).

5. Some discussion on the inclusion and impact of negative variants on ML prediction of combinatorial variants and mutational trajectories would be very helpful.

We agree with the suggestion and have added the following paragraph in the ML section (Page 6 in the manuscript; Figure S6 in the SI):

Finally, to assess the quality of L10, we compared the ratio between positive (FIOWT ≥ 1) and negative variants (FIOWT < 1). While library L2 to L7 predominantly displayed negative variants (ranging from 66 % to 97 %), the filtered ML library L10 enabled the enrichment of positive variants (76 %; **Figure S6**). As we provided all collected data (e.g., sequence-function data of positive and negative variants) to train the Gaussian process and subsequently filtered with prior knowledge, it is unclear if the increase in the fraction of hit enzymes was achieved by learning to build on the positive combinations or by learning to avoid the unfavorable combinations, or both. In this sense, we recommend that all data points collected under comparable experimental conditions, including data connected to negative variants, are provided to improve on the predictive ability of machine learning models.

Library	sites targeted	# positive variants (FIOWT ≥ 1)	# negative variants (FIOWT < 1)	# total variants	% positive variants (FIOWT ≥ 1)	% negative variants (FIOWT < 1)
2	97, 174, 238, 241, 242, 245	39	76	115	33.9	66.1
3	174-242-245	15	435	450	3.3	96.7
4	241-242-245	92	588	680	13.5	86.5
5	238-242-245	68	372	440	15.5	84.5
6	174-238-241	163	400	563	29	71
7	174-238-241-242-245	37	725	762	4.9	95.1
10 (filtered - ML)	(174-238-241-242-245)	261	83	344	75.9	24.1

Figure S6. Distribution of “positive” (FIOWT ≥ 1) and “negative” (FIOWT < 1) variants screened in libraries L2 – L7 as well as in the ML-filtered library L10. Unique on-target variants are considered for libraries L2 – L7 as sequence-function results were available for these datapoints. In case of L10, the total number of screened transformants is considered as only the hits were sequenced. For L10, predicted variants with mutations on the 5 sites were obtained, followed by a filtering process according to criteria specified in Table S14.

While the abstract mentions modeling and kinetic studies, providing a bit more information on the key findings from these studies could enhance the understanding of the mechanistic factors governing the improved reaction outcome.

To enhance the informativeness of the Abstract, we have enumerated the mutations that exert the most significant impact on activity. More findings are provided and explained in a greater detail within the Enzyme kinetics and Enzyme modelling sections of the manuscript.

Some minor typos or corrections include:

Strictly speaking paper sometimes suffers from unclear language or imprecise statements. For instance, these sentences in the abstract have to be:

“While the improvement of protein structure, stability and solubility has been frequently explored in this way, enzyme activity has proven to be a more difficult target function.” → it means improvement of prediction of protein structure, stability, and solubility ... → as we cannot improve the protein structure rather its prediction, sentence is correct for the stability and solubility. Additionally, a comprehensive review in this field (<https://doi.org/10.1021/acscatal.3c02743>) shows several examples of ML- guided engineering of enzyme activity. Which contradicts the statement made by authors that “enzyme activity has proven to be a more difficult target function”.

As suggested, we have removed the sentence from the abstract. In addition, the suggested Review paper has been cited in the answer to Comment No. 2.

“Harnessing the power of algorithm-assisted enzyme engineering, we created a 10-amino acid variant exhibiting a 64-fold higher apparent k_{cat} and improved robustness under process conditions compared to the wild-type enzyme.” → 10-amino acid variant is misleading → it should be described as e.g. 10-amino acid substituted variant or a variant harboring 10 substitutions.

Thanks for this suggestion. We changed to “10-amino acid substituted variant”.

-Line 65, references 3233 should be either comma or dash separated.

References have been properly grouped to avoid any confusion.

REVIEWERS' COMMENTS:

Reviewer #1 (Remarks to the Author):

All comments have been addressed. The manuscript is great and ready to publish. Congratulations, this is excellent enzyme engineering work.

Reviewer #3 (Remarks to the Author):

I would like to express my sincere appreciation to the authors for their diligent efforts in addressing my comments. The revisions made have significantly strengthened the paper, demonstrating a clear commitment to enhancing its quality. The authors have thoughtfully incorporated the suggested changes, providing thorough explanations where necessary. The revised manuscript now aligns well with the standards of the journal.

Reviewer #1 (Remarks to the Author):

All comments have been addressed. The manuscript is great and ready to publish. Congratulations, this is excellent enzyme engineering work.

We are grateful to Reviewer 1 for his/her positive feedback and the careful reviewing our manuscript. The interaction has been a pleasure.

Reviewer #3 (Remarks to the Author):

I would like to express my sincere appreciation to the authors for their diligent efforts in addressing my comments. The revisions made have significantly strengthened the paper, demonstrating a clear commitment to enhancing its quality. The authors have thoughtfully incorporated the suggested changes, providing thorough explanations where necessary. The revised manuscript now aligns well with the standards of the journal.

We appreciate the valuable input of Reviewer 3 which has led to an improved manuscript. Many thanks for his/her positive feedback on the final version.